# Continuous structure modification of metal-organic framework glasses via halide salts

Fengming Cao [1,9], Søren S. Sørensen [1,9] ✉, Anders K. R. Christensen[1], Samraj Mollick[1], Xuan Ge[1], Daming Sun[1], Anders B. Nielsen[2,3], Niels Chr. Nielsen [2,3], Nina Lock[3,4], Ronghui Lu[3,4], Rebekka Klemmt [3,5], Peter K. Kristensen[6], Lars R. Jensen [6], Francesco Dallari [7], Jacopo Baglioni [7], Giulio Monaco [7], Martin A. Karlsen [8], Volodymyr Baran [8] & Morten M. Smedskjaer [1] ✉

Melting and glass formation of metal-organic frameworks (MOFs) allow them to be processed into bulk materials. However, two major challenges remain: only a small fraction of MOF crystals undergo melting and glass-formation, and no well-established strategies exist for tuning MOF glass structures and properties. Here, we address both challenges through co-melting of zeolitic imidazole frameworks (ZIFs), a subset of MOFs, with heterocycle-based halide salts. The salt acts as a chemical "modifier", akin to the role of alkali modifiers in traditional silicate glasses, e.g., allowing the melting of ZIF-8 that otherwise decomposes prior to melting. Through experimental and computational analyses, we show that the salts depolymerize the ZIFs, enabling continuous tuning of the fraction of bridging to non-bridging imidazolate linkers and, thereby, the thermal and mechanical properties. The proposed strategy enables diversification of MOF glass chemistry, tunable structures and properties, and ultimately an increased number of glass-forming MOFs with improved functionalities.

Inorganic and organic glass materials exhibit remarkable versatility in processing, with their abilities to be shaped into various forms and for their chemical compositions to be continuously tuned[1]. An important family of organic-inorganic hybrid materials, metal-organic frameworks (MOFs), have not yet achieved the same level of processability and continuous composition tunability[2]. MOFs offer exciting prospects for their uses as, e.g., catalysts[3], gas adsorbers[4,5], and electrolytes[6], especially enabled by their high porosity (surface area up to >10,000 m² g⁻¹)[4,7,8], but their practical applications are often hindered by poor processability as they are typically only available in powder form[9].

The recent discovery of MOF glasses through melting and quenching may help solve this problem as they maintain many properties of the parent MOF crystals while also allowing the processing into grain-boundary-free bulk (>1 cm) objects through hot forming techniques[10]. However, to realize processable MOFs, a key challenge is to tune the viscosity–temperature curve, which is essential for shaping MOF glasses into desired shapes and configurations, beyond mere melting or sintering techniques[11]. Another significant hurdle is the limited number of meltable MOF crystals, as the melting temperature of many MOFs is higher than their decomposition temperature ($T_m > T_d$)[12]. This restricts the continuous composition tuning of MOF

[1]Department of Chemistry and Bioscience, Aalborg University, Aalborg, Denmark. [2]Department of Chemistry, Aarhus University, Aarhus, Denmark. [3]Interdisciplinary Nanoscience Center (iNANO), Aarhus University, Aarhus, Denmark. [4]Department of Biological and Chemical Engineering, Aarhus University, Aarhus N, Denmark. [5]iMAT Aarhus University Centre for Integrated Materials Research, Aarhus University, Aarhus, Denmark. [6]Department of Materials and Production, Aalborg University, Aalborg, Denmark. [7]Department of Physics and Astronomy 'Galileo Galilei', University of Padova, Padova, Italy. [8]Deutsches-Elektronen Synchrotron (DESY), Hamburg, Germany. [9]These authors contributed equally: Fengming Cao, Søren S. Sørensen. ✉e-mail: soe@bio.aau.dk; mos@bio.aau.dk

glasses. Currently, the number of known MOF glasses is <200, in stark contrast to the >100,000 known MOF crystals[13,14]. In comparison, there are more than 200,000 known oxide glass compositions in the Sci-Glass database, compared to ~70,000 oxide crystals[15]. This difference is ascribed to the inherent feature of oxide glasses to be formed in a nearly infinite number of compositions through mixing of pure glass network formers ($SiO_2$, $GeO_2$, $B_2O_3$, etc.) with so-called "modifiers" (e.g., alkali, alkaline earth, and transition metal oxides)[16]. These modifiers typically depolymerize the oxide network structure, allowing continuous modification of glass structure and properties. Despite the success of using modifiers in oxide as well as sulfide, fluoride, and chalcogenide glasses[1], no well-established modification platform exists for MOF glasses.

Several families of glass-forming MOFs exist, including zeolitic imidazolate frameworks (ZIFs)[17]. ZIFs feature topologies analogous to those of silica or zeolite networks (Fig. 1a), with tetrahedra consisting of a metal ion (e.g., $Co^{2+}$, $Zn^{2+}$) and four imidazolate-based ligands[14,18]. The first family of MOFs with demonstrated melting behavior was ZIFs and they have since then been the most studied MOF glass family. This includes the discovery of various meltable compositions (e.g., ZIF-4, ZIF-62, ZIF-UC-5, and TIF-4)[19,20], simulation studies of the bond-breaking behavior leading to melting and glass-formation[21], as well as the effect of pressure and exchange of linkers[22] on phase transitions[23]. More recent studies have investigated the effect of changing the metal node[24,25] and organic linker ratio[24], as well as the addition of water[26] and an ionic liquid[12] on various types of hybrid organic–inorganic glasses. In ZIF-62 ($Zn(Im)_{1-x}(bIm)_x$, where Im is imidazolate and bIm is benzimidazolate), changing the linker ratio results in only moderate changes of $T_m$ and $T_g$ (by about 70 and 30 °C, respectively, when $x$ changes from 0.35 to 0.03). Hydrothermal treatment of ZIF-62 glass gives rise

to $OH^-$ incorporation and results in larger changes of $T_m$ (from 447 to <300 °C) and $T_g$ (from 320 to ~200 °C)[26]. Another study showed that ionic liquid (1-ethyl-3-methylimidazolium bis(trifluoromethane sulfonyl)imide) addition enables melting of the otherwise unmeltable ZIF-8 ($Zn(mIm)_2$, where mIm is 2-methylimidazolate) crystal, yet with transition temperatures comparable to those of other meltable ZIFs ($T_m \approx 380$ °C and $T_g \approx 320$ °C)[12].

The concept we attempt to realize herein is to break up the polymerized ZIF network by introducing new types of bonds in the system. Inspired by the concepts of bridging and non-bridging oxygens in oxide glasses, mixed linker systems in, e.g., oxysulfides and oxynitrides (where oxygen, sulfur, and nitrogen link metal centers)[1], and a crystalline Zn-imidazolate-chloride system[27] (where both imidazole and Cl bond to Zn), we propose a method for modifying ZIF glasses by adding a halide salt. Specifically, we use heterocycle-based halide (Cl, Br, I) salts for co-melting with ZIF crystals, achieving melting below 300 °C, very low $T_g$ values (<150 °C), and the preparation of bubble-free bulk-sized samples (>1 cm). We chose these salts as they mimic the linker polarity of the existing framework, and they introduce halide ions, which are known to allow for ligand exchange in ZIF-like networks[28]. Through ex situ and in situ structural analyses, molecular dynamics simulations based on density functional theory (DFT), and mechanical and thermal characterization, we conclude that the halide salts act as modifiers in the ZIF network by partially replacing the imidazole-based organic linkers with halide anions (Fig. 1a). This method provides a pathway for continuously tuning, e.g., glass transition temperature, mechanics, and chemical durability of MOF glasses, and ultimately a means to increase the diversity of synthesizable MOF glasses.

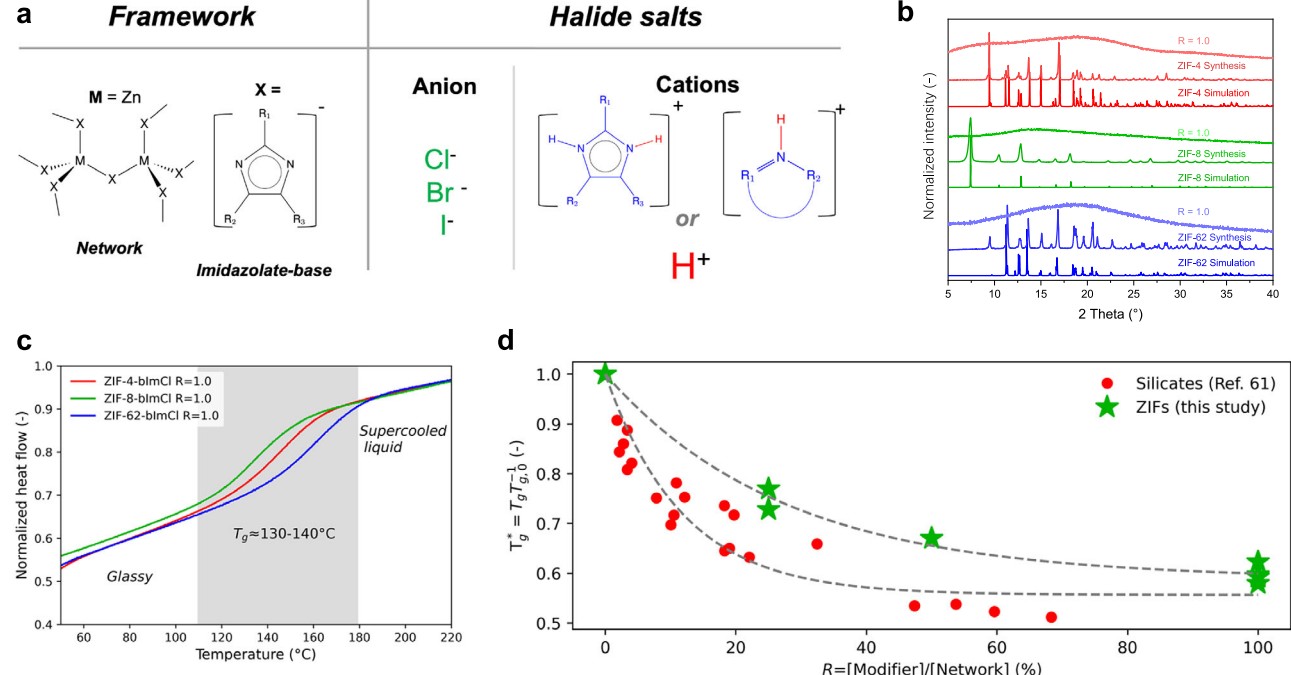

**Fig. 1 | Structural, crystallographic, and thermal properties of modified zeolitic imidazolate frameworks. a** Sketch of polymerized zeolitic imidazole framework and its imidazolate-based linkers (left) and the added halide salts (right). **b** X-ray diffraction patterns of ZIF-4, ZIF-8, and ZIF-62, including simulated (CCDC deposition numbers: ZIF-4:602538, ZIF-8:1429243, and ZIF-62:671070) and experimental crystalline forms and experimentally formed glasses upon co-melting with benzimidazolium chloride (R = 1.0). **c** Fast scanning calorimetry heat flow data of ZIF-4, ZIF-8, and ZIF-62 co-melted with benzimidazolium chloride (R = 1.0), showing glass transition temperatures ($T_g$) in the range of 130-140 °C. The

heating rates used for these measurements were 500 K s⁻¹. We note that heat flow data is normalized to the maximum heat flow due to the very small sample sizes (micro or nanogram), making it infeasible to measure sample masses. **d** Effect of modifier content on the reduced glass transition temperature ($T_g^* = T_g T_{g,0}^{-1}$, where $T_g$ is the glass transition of a modified glass and $T_{g,0}$ is the glass transition temperature of a pure glass former), shown for all modified ZIF-4 and ZIF-62 glasses of the present study (see Supplementary Table 1) as well as literature data for silicates[61]. The dashed lines represent guides for the eye based on exponential decay functions. Source data are provided as a Source Data file.

## Results

### Preparation and properties of modified ZIF-derived glasses

Our study focuses on the well-known ZIF-62 [Zn(Im)$_{1.75}$(bIm)$_{0.25}$], ZIF-4 [Zn(Im)$_2$], and ZIF-8 [Zn(mIm)$_2$] (see Methods). ZIF-62 and ZIF-4 are known glass-formers with melting temperatures of 430 and 593 °C, respectively[19]. In contrast, ZIF-8 is known to decompose before melting ($T_d$ ~450 °C)[12]. We mix each of these three ZIF crystals with varying amounts ($R = 0.25, 0.5, 0.75, 1.0$) of heterocyclic halide salts (Fig. 1a), where $R$ is the molar ratio between the added amounts of ZIF crystal and modifier salt, i.e., $R = [n_{modifier}]/[n_{ZIF}]$. We note that we mainly investigate the glasses with added benzimidazolium chloride salt at a modifier ratio of $R = 1.0$ due to their lower hygroscopicity and thereby easier handling for all testing. However, to demonstrate the universality of our approach, we perform selected thermal, mechanical, and structural analyses for samples with varying $R$ values and halide ions.

Figure 1b shows the X-ray diffractograms of crystalline ZIF-4, ZIF-8, and ZIF-62, as well as these crystals co-melted with benzimidazolium chloride (H$_2$bImCl) for $R = 1.0$ at a temperature of 300 °C. All these systems become non-crystalline upon mixing, heating, and quenching. We note how this melting temperature is below that of most other known ZIF systems, for example, ZIF-UC-6[29] [Zn(Im)$_{1.82}$(5-aminobenzimidazolate)$_{0.18}$] ($T_m = 345$ °C, $T_g = 316$ °C), which melts at a markedly lower temperature than other Pbca ZIFs, linker-functionalized TIF-4 [Zn-(Im)$_{1.8}$(mbIm-5-methylbenzimidazolate-C$_8$H$_7$N$_2$)$_{0.2}$] ($T_m = 440$ °C, $T_g = 336$ °C), and ZIF-UC-5 [Zn(Im)$_{1.8}$(ClbIm-5-chlorobenzimidazolate-C$_7$H$_4$N$_2$-Cl$^-$)$_{0.2}$] ($T_m = 428$ °C)[30]. After melt-quenching, the recovered samples were subjected to thermal analyses through fast differential scanning calorimetry (FDSC) measurements (Fig. 1c). Using FDSC, we identify clear glass transition temperatures ($T_g$) for all three ZIFs in the range of 130–140 °C (see Supplementary Table 1), i.e., ~200 °C below that of the $T_g$ of the unmodified glasses in the case of ZIF-4 and ZIF-62 as measured by traditional DSC (note that ZIF-8 does not melt at $R = 0$ under standard conditions)[19,31]. This confirms the formation of glasses for all the co-melted samples. To further validate the glass transition behavior observed through FDSC, we have performed DSC measurements at a conventional heating rate of 10 K min⁻¹. As shown in Supplementary Figs. 1 and 2, these DSC data confirm that the modified ZIF-derived glasses can be prepared even at standard cooling rates (10 K min⁻¹), exhibiting clear glass transitions that are qualitatively consistent with the FDSC results. We note how the heat capacity jumps ($\Delta C_p$) across the glass transition, as quantified using standard DSC data, are higher for the modified compared to the pure ZIF-4 and ZIF-62 glasses. The absolute $T_g$ values are lower in the standard DSC measurements compared to those observed using FDSC due to the difference in heating rate (and its relation to liquid fragility)[32]. Importantly, no additional thermal events, such as ligand decomposition or phase separation, are detected during the lower heating rate experiments, confirming the thermal stability of the modified glasses.

Addition of modifiers in network glasses, e.g., silicates, is generally associated with depolymerization[1]. This results in a monotonic change of glass properties with the modifier content, such as decreasing $T_g$ with increasing modifier content[1]. To study if this is also the case for the present samples, we have prepared glassy mixtures of each of the three ZIFs co-melted with H$_2$bImCl for $R$ values of 0.25, 0.5, 0.75, and 1.0 (see Supplementary Figs. 3, 4). Generally, higher temperatures are required to fully melt the system for lower values of $R$ (see XRD results in Supplementary Fig. 3), but we note that all samples reach a fully molten state at ~300–360 °C (see photographs of recovered samples in Supplementary Fig. 4). In contrast, the corresponding mixing at room temperature did not result in any reaction (Supplementary Fig. 5). Using both FDSC and standard DSC, we find a monotonic decrease in $T_g$ with increasing modifier content, in good resemblance with other network glasses (see comparison of effect of varying $R$ on $T_g$ in Fig. 1c,

d and raw data in Supplementary Figs. 1 and 6, and Supplementary Table 1). In addition, we observe that the modified glasses are prone to a permanent increase in $T_g$ upon subsequent heating (>300 °C) for few minutes or even seconds (Supplementary Fig. 7). An effect we assign to evaporation of Im and bIm species coming from both the network and added modifier salt (see Supplementary Text). Considering the mechanical properties, the Vicker's hardness of the ZIF-4-bImCl and ZIF-62-bImCl ($R = 1.0$) glasses is lower (~0.35 GPa, Supplementary Fig. 8a) than that of the unmodified ZIF glasses (~0.6 GPa)[33,34]. Similarly, the crack initiation resistance[32] of ZIF-4-bImCl and ZIF-62-bImCl ($R = 1.0$) glasses are 0.12 N and 0.38 N, respectively (Supplementary Fig. 8b), significantly lower than that of the unmodified ZIF-62 glass (~2 N)[35].

Photographs and scanning electron microscopy (SEM) images of the samples are shown in Fig. 2a, d and Supplementary Fig. 9. The samples are found to take the shape of the container in which they were melted and feature smooth surfaces, indicating viscous flow and a grain-boundary-free glassy state. Furthermore, the modified ZIF-4 and ZIF-62 (with H$_2$bImCl at a $R = 1.0$ ratio) samples are transparent (Supplementary Fig. 10). We note that these measurements on ZIF-8-derived glasses were not performed due to their poor optical transparency. We find it possible to produce defect-free samples with size of >1 cm for the modified ZIF-4, ZIF-8, and ZIF-62 glasses (Fig. 2a). This stands in significant contrast to the case of unmodified ZIF-62 and especially unmodified ZIF-4 glass, the quality of which are often impaired by partial decomposition and foaming[10] (Fig. 2b, c). Furthermore, previous efforts to lower the melting point of ZIF-8 below its decomposition temperature using ionic liquids, which employed melting temperatures around 600 °C, created samples with sizes of <1 mm[12]. Based on the observed transparency for ZIF-4 and ZIF-62 derived glasses and the compositional analysis performed using energy-dispersive X-ray (EDX) spectroscopy (Fig. 2e-h, Supplementary Fig. 9, Supplementary Table 3), we infer that the samples are homogenous in the elements tested (C, N, Zn, Cl) at both the micro- and macro-scales. However, we note that the reported compositions are approximate due to difficulties in data analysis arising from overlapping signals, especially for carbon and nitrogen (see Supplementary Text).

In addition to using H$_2$bImCl as the modifier salt, we also find glass formation upon co-melting the ZIF crystals using benzimidazolium salts with bromide (Br⁻) and iodide (I⁻) as anions. In summary, we find similar $T_g$ values and loss of crystallinity upon melt-quenching for the Br⁻ and I⁻ based salts as for the Cl⁻ based salts (see XRD and calorimetry data as well as photographs in Supplementary Figs. 1 and 11–13), indicating a similar mechanism of ZIF modification.

Finally, we have performed CO$_2$ adsorption measurements on selected samples at 22, 0, and −42 °C to characterize the CO$_2$ gas uptake capabilities of the modified glasses, as preliminary N$_2$ adsorption measurements at 77 K yielded negligible or unmeaningful adsorption isotherms[36]. As shown in Supplementary Fig. 14, we find that the CO$_2$ adsorption capacity increases as temperature decreases, following fundamental thermodynamic principles. While pure ZIF-4 and ZIF-62 glasses exhibit CO$_2$ uptakes of a few mmol CO$_2$ per gram of glass, similarly to previously reported values[37], benzimidazolium chloride (bImCl) modification at $R = 1.0$ significantly reduces the adsorption capacity. In contrast, ZIF-8-bImCl ($R = 1.0$) glass shows higher CO$_2$ uptake than the modified ZIF-4 and ZIF-62 glasses, suggesting framework-dependent effects. Partial modification ($R = 0.25$) in ZIF-62 derived glass retains some adsorption capacity compared to full modification ($R = 1.0$). Additionally, ZIF-62-bImI ($R = 1.0$) glass exhibits enhanced CO$_2$ uptake at lower temperatures compared to ZIF-62-bImCl ($R = 1.0$), highlighting the influence of halide substituents.

### Structural characterization

Having established the thermodynamical and mechanical characteristics of the modified ZIF-derived glasses, we next investigate their

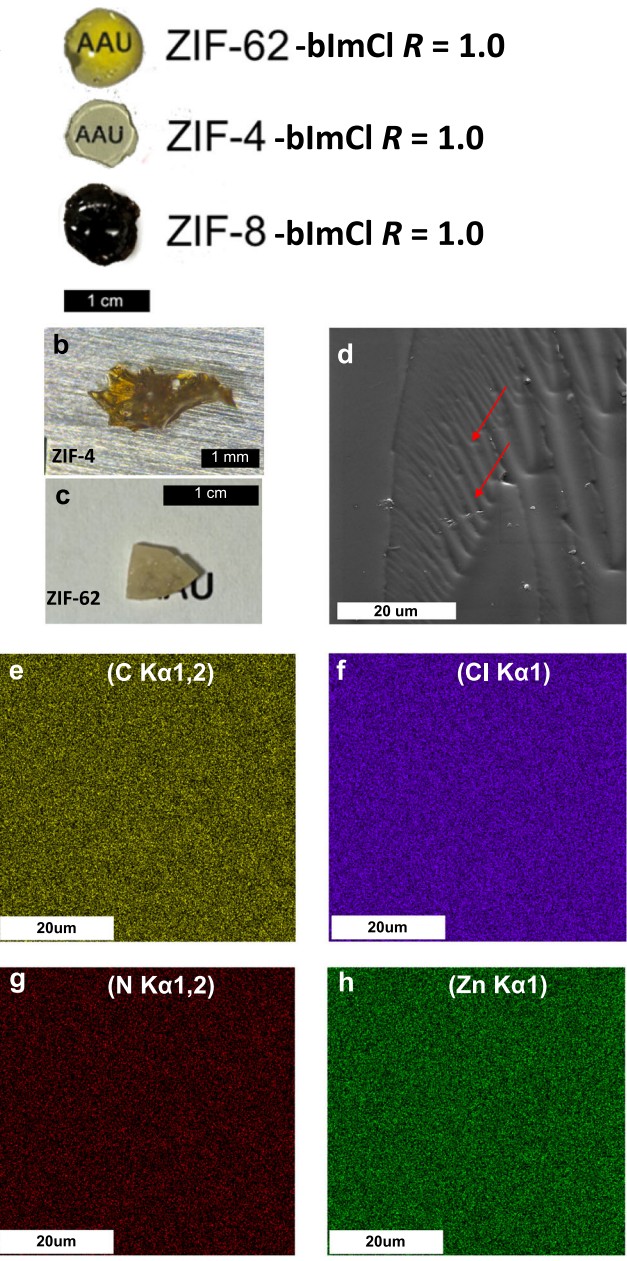

**Fig. 2 | Photographs, SEM images, and compositional mapping of H₂bImCl modified ZIF-derived glasses. a** Photographs of bImCl-modified ZIF-4, ZIF-8, and ZIF-62 ($R = 1.0$) glasses after melt-quenching. Clear transparency is observed for the ZIF-4-bImCl and ZIF-62-bImCl samples. **b**, **c** Photographs of unmodified: **b** ZIF-4 and **c**, ZIF-62 glasses. Note the scale differences in (**a**–**c**). **d** SEM image of a ZIF-4-bImCl ($R = 1.0$) glass, showing smooth viscous flow and fracture lines, suggesting the existence of a liquid-like state during forming (red arrows). EDX measurements of ZIF-4-bImCl $R = 1.0$ glass for **e** C, **f** Cl, **g** N, and **h** Zn elements.

structural features. Two previous studies have solved the single-crystal structure of a bis(imidazole)zinc(II) chloride phase, revealing a mixed linker structure, where each tetrahedral Zn is coordinating two imidazoles and two chloride anions[27,28]. Another very recent work prepared closely-related bis(benzimidazole)zinc(II) halide (Cl, Br, I) single crystals and showed that these can be melt-quenched into glasses with $T_g$ values around 80 °C[38]. In the following, we demonstrate that the present co-melted ZIF-modifier systems adopt largely similar local configurations, where the Zn nodes coordinate to a mixture of halide anions and imidazolate-based linkers. However, our approach is fundamentally different in that it does not require the prior preparation of

a phase-pure single-crystalline phase, i.e., it is akin to network modification in traditional glasses, as the present samples can feature any fraction of bridging to non-bridging imidazolate linkers. Consequently, this enables continuous tuning of both structure and properties in this material family.

We first analyze the synthesized ZIF-4-bImCl, ZIF-62-bImCl, and ZIF-8-bImCl samples (with varying $R$) using Fourier transform infrared (FTIR) and Raman spectroscopy, revealing that the imidazolate and benzimidazolate linkers remain intact and do not decompose upon melt-quenching (Supplementary Figs. 15, 16, Supplementary Table 2, Supplementary Text). To further probe the structural modifications induced by halide incorporation, we have performed far-infrared (FIR) spectroscopy measurements. As shown in Supplementary Fig. 17, we identify a peak at about 310 cm⁻¹, which is ascribed to the Zn–N stretching mode[39] as well as a mode most likely associated with a Zn-halide bond (~280 cm⁻¹). We also probe the structure of the modified ZIF-derived glasses (for $R = 1.0$ using bImCl) through X-ray photoelectron spectroscopy (XPS, see Supplementary Figs. 18-19) and compare the spectra to those of the unmodified ZIF crystals[40] (noting that the XPS signal of ZIF glasses and crystals are known to be very similar[40]). Upon glass formation, we observe a broadening of the C 1$s$, N 1$s$, and Zn 2$p$ peaks as well as an additional shoulder peak in the N 1$s$ spectra for all glasses at relatively high energy (~404 eV). We assign the latter to the formation of pyrrolic nitrogen (N bonded to both C and H) upon co-melting with the halide salt, as this species is introduced through the benzimidazolium cation in contrast to the lower-energy peak (~402 eV) from pyridinic nitrogen (N bonded to C). We note that the C 1$s$ XPS peaks (see Supplementary Fig. 19a, d, g) in bImCl-incorporated ZIF-derived glasses shift to lower binding energy, indicating increased electron density around carbon atoms. This shift is attributed to π-electron-rich benzimidazole or imidazole species, which polarize the local environment and enhance electron density within the carbon framework[41].

To further assess the incorporation and integrity of benzimidazole in the modified ZIF-derived glasses, we have performed liquid-state ¹H nuclear magnetic resonance (NMR) spectroscopy measurements on digested samples of ZIF-4-bImCl, ZIF-8-bImCl, and ZIF-62-bImCl for $R = 1.0$ (Supplementary Fig. 20 and Supplementary Table 4). The proton exchange between benzimidazolium and imidazolate linkers suggests that benzimidazolium cations can replace existing organic linkers by being transformed to benzimidazole. Importantly, the retention of characteristic benzimidazole and imidazole ¹H NMR signals in solution confirms that no significant linker decomposition occurs during the glass formation process.

We then characterize the structure of the modified ZIF-62-bImCl ($R = 1.0$) glass using high-field solid-state ⁶⁷Zn, ¹⁵N, ¹³C, and ¹H magic-angle spinning (MAS) NMR spectroscopy and compare the results to those of the unmodified ZIF-62 glass[26,28]. The ⁶⁷Zn MAS NMR data in Fig. 3a (including a simulated spectrum made using SIMPSON[42]) show a slight change in the chemical shift upon co-melting with the H₂bImCl salt. In comparison to previous ⁶⁷Zn measurements of ZIF-62 glass ($\delta_{iso} = 278$ ppm)[43] and crystalline bis(imidazole)zinc(II) chloride ($\delta_{iso} = 265$ ppm)[27], the present glass features a chemical shift of $\delta_{iso} = 268$ ppm, indicating a bonding environment very similar to that in bis(imidazole)zinc(II) chloride crystal, i.e., suggesting direct bonding between Zn and Cl⁻.

Furthermore, as shown in Supplementary Figs. 21–23, we find significant changes in the solid-state ¹⁵N MAS NMR shifts upon halide salt addition, including the appearance of multiple new peaks, which we assign to non-bridging benzimidazole and imidazole species as introduced from the H₂bImCl salt and the network itself. This is because the two N are not equivalent in the modified ZIF glass structure as one is protonated ($\delta < 180$ ppm) while the other is linked to Zn ($\delta > 180$ ppm)[44,45] (see sketches in Supplementary Figs. 21–23). Furthermore, the addition of benzimidazole species through the H₂bImCl

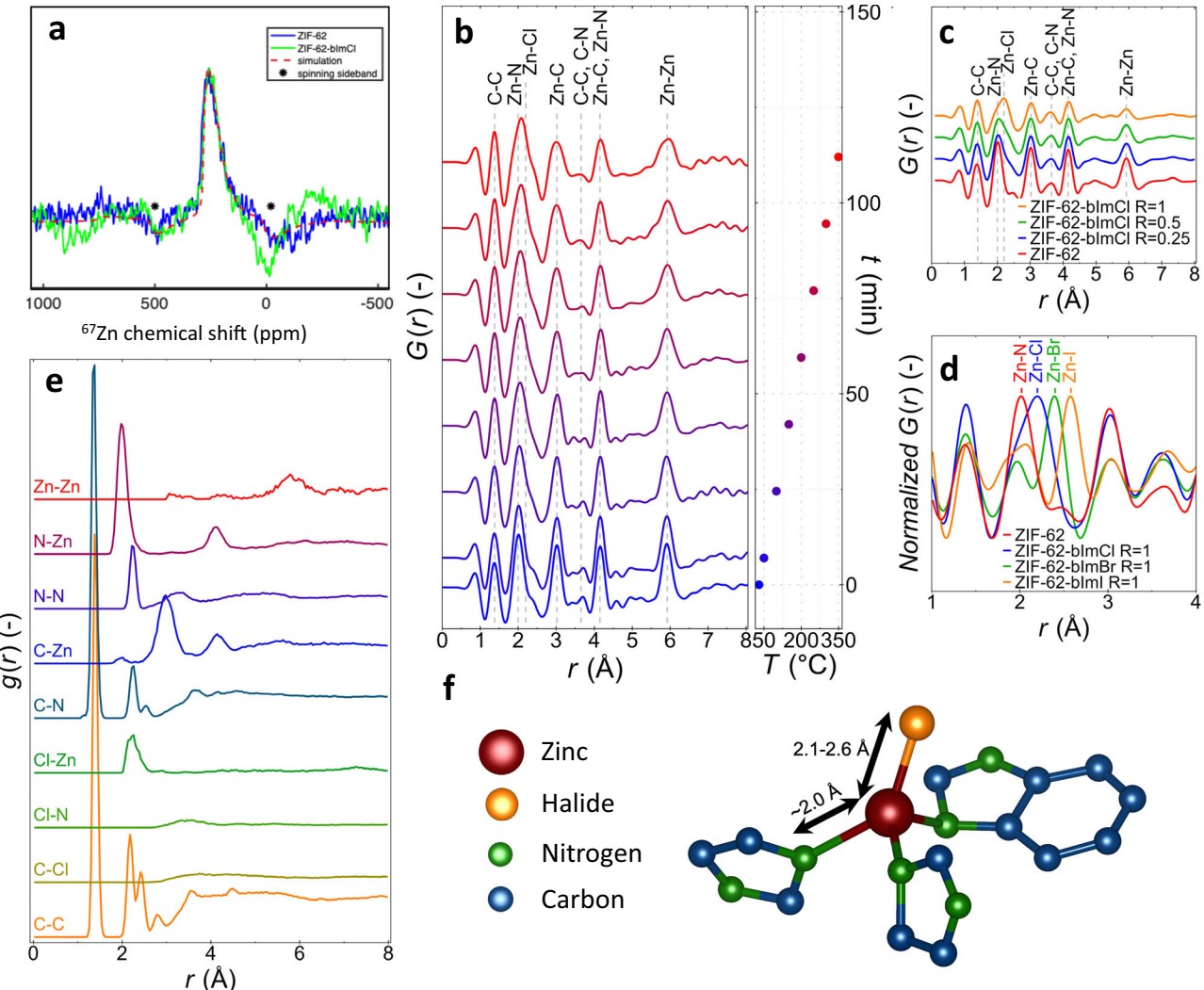

**Fig. 3 | Structural analysis of modified ZIF-derived glasses using $^{67}$Zn solid-state NMR and in situ pair distribution function analysis. a** $^{67}$Zn MAS NMR spectrum of unmodified ZIF-62 and modified ZIF-62-bImCl ($R = 1.0$) glasses. Spinning sidebands are denoted with '*'. **b** In situ variable temperature pair distribution function (PDF), $G(r)$, of the ZIF-62-bImCl $R = 0.5$ sample, starting from the mixture of ZIF crystal and $H_2$bImCl salt at time $t = 0$. **c**, PDFs of glassy ZIF-62-bImCl samples with $R = 0, 0.25, 0.5$, and $1.0$ (at room temperature) with assigned peaks. We note how C–C and C–N peaks stem from the internal correlations of both imidazolate and benzimidazolate linkers. **d** Example of the 1-4 Å region of the PDF (normalized by maximum intensity) of ZIF-62-bImB glasses where B = [Cl, Br, I]. All PDFs shown in (**c**, **d**) have been modified using a Lorch function with $Q_{max} = 20$ Å$^{-1}$. **e** Partial PDFs of a simulated structure of ZIF-62-bImCl $R = 0.5$. **f** Atomic snapshot from first principles simulation of the structure of a Zn tetrahedron where one imidazolate linker is exchanged for a halide. Source data are provided as a Source Data file.

salt is clearly seen as an increase in the intensity at $\delta \sim 190$ ppm (Supplementary Fig. 21). Similarly, the solid-state $^{13}$C MAS NMR measurements show the enhanced intensity of peaks due to benzimidazole species in the structure ($\delta \sim 110-130$ ppm region in Supplementary Fig. 22). New peaks at $\delta \sim 135$ ppm furthermore indicate the asymmetry of electron density associated with the pure imidazole and benzimidazole species[46]. The solid-state $^1$H MAS NMR measurements reveal a broad peak around 5–7 ppm for both ZIF-62 and ZIF-62-bImCl glasses, which we assign to the aromatic proton. Finally, an additional peak at -12 ppm is observed in the ZIF-62-bImCl glass (Supplementary Fig. 23), which we ascribe to the N-H proton in pure imidazole and benzimidazole species[46].

To confirm the presence of mixed organic-halide linkers in the modified ZIF-derived glasses, we have performed in situ high-temperature X-ray pair distribution function (PDF) measurements of the three crystalline ZIFs mixed with the imidazolium-based halide salts (Cl⁻, Br⁻, I⁻). An example of the recorded $G(r)$ data is shown in Fig. 3b for the ZIF-62-bImCl ($R = 0.5$) sample heated from room temperature to 300 °C and then cooled back to room temperature. The

reciprocal space data (total scattering structure factor, $S(Q)$) are shown in Supplementary Fig. 24a, revealing a loss of crystallinity at around ~300 °C, indicating melting at a temperature much lower than the $T_m$ of unmodified ZIF-62 (~450 °C). The PDF data in Fig. 3b mainly show changes in the intensities and shape of the peaks around 2, 4, and 6 Å, which all appear to broaden with increasing temperature.

We also present room temperature PDF data of the melt-quenched ZIF-62-bImCl glasses with $R$ values of 0, 0.25, 0.5, and 1.0 (Fig. 3c). We note that these samples were cooled faster (in our laboratory) than what was possible during the in situ PDF experiments at the beamline. We find that the ZIF-62-bImCl glass features both new peaks and differences in peak intensities in comparison to the unmodified ZIF-62 glass. For example, the peak at ~2.1 Å broadens and shifts toward a higher interatomic distance, consistent with the expected Zn-Cl bond (~2.2 Å) (see Fig. 3c)[27,47] compared to the existing Zn-N bond (~2.0 Å)[31]. The reciprocal space data (total scattering structure factor, $S(Q)$) for Fig. 3c, d are shown in Supplementary Fig. 24b. Similarly, we observe a broadening and decrease of intensity of the peak at ~6 Å, corresponding to the Zn−Zn correlation, thus indicating a reduction in

the number of bridging imidazolate species (i.e., Zn-Im-Zn), which we again interpret as being due to the incorporation of Cl. These findings support a mechanism of depolymerization of the ZIF network through partial (benz)imidazolate linker exchange for Cl. We thus note how the distinction between a ZIF glass and a coordination polymer glass becomes less clear in the modified non-crystalline state. However, while the Zn–Zn correlation weakens with increasing modifier concentration, its persistence across all samples confirms the presence of linked Zn-polyhedra, maintaining a fundamental resemblance to the ZIF structure.

We have also probed the structure of the modified Br and I analogous glasses (Fig. 3d). We find strong peaks corresponding to the expected bond lengths of Zn-Br (~2.4 Å) and Zn-I bonds (~2.6 Å), supporting the partial (benz)imidazolate for halide exchange[48,49]. The in situ PDF data for these systems show that the Zn-Br and Zn-I bonds are formed upon heating (Supplementary Figs. 25, 26). This is more evident in the Br⁻ and I⁻ glasses compared to the Cl⁻ glass due to the longer bond lengths of Zn-Br and Zn-I compared to Zn-N, while the bond lengths of Zn-Cl and Zn-N are more similar. The appearance of a Zn-halide bond is found in all studied systems containing halide ions as compared to the pure ZIFs (Supplementary Figs. 27–29). This includes ZIF-4-bImCl (Supplementary Fig. 30), ZIF-8-bImCl (Supplementary Fig. 31), ZIF-62-bImCl (with varying R, Supplementary Figs. 32–35), ZIF-

62-bImBr (Supplementary Fig. 25), and ZIF-62-bImI (Supplementary Fig. 26). Additionally, these systems exhibit varying amorphization/melting temperatures (200–350 °C) that decrease with increasing amount of halide modifier salt. X-ray PDF measurements further reveal that Zn-halide bonds start forming at around 100 °C when heating the ZIF network with the modifier salt. We have also tested this effect using calorimetry for the ZIF-62-bImCl $R = 1.0$ sample in an open crucible (Supplementary Fig. 36). We first observed a small endothermic peak at around 100 °C, likely associated with the melting of the modifier salt, followed by a broad endotherm in the 200–250 °C range. We argue that this likely indicates co-melting of the ZIF framework in the molten salt matrix, which is similar to how network "formers" in oxide glasses (e.g., $SiO_2$) can be flux-melted at temperatures below their own thermodynamic melting point[1]. To further test the universality of our modifier approach, we have also used imidazolium chloride (ImCl) and pyridinium chloride (PyCl) as modifiers and co-melted these with the different ZIF crystals. We find that these chloride compounds also lead to transparent bulk solids (except for ZIF-4-PyCl, which is transparent but liquid-like). The ZIF-62-ImCl system was also tested by in situ X-ray diffraction, as shown in Supplementary Fig. 37. All these samples were found to be non-crystalline (Supplementary Figs. 38, 39).

In addition to the above experiments, we have performed DFT-based molecular dynamics (DFT-MD) simulations of the ZIF-62-bImCl

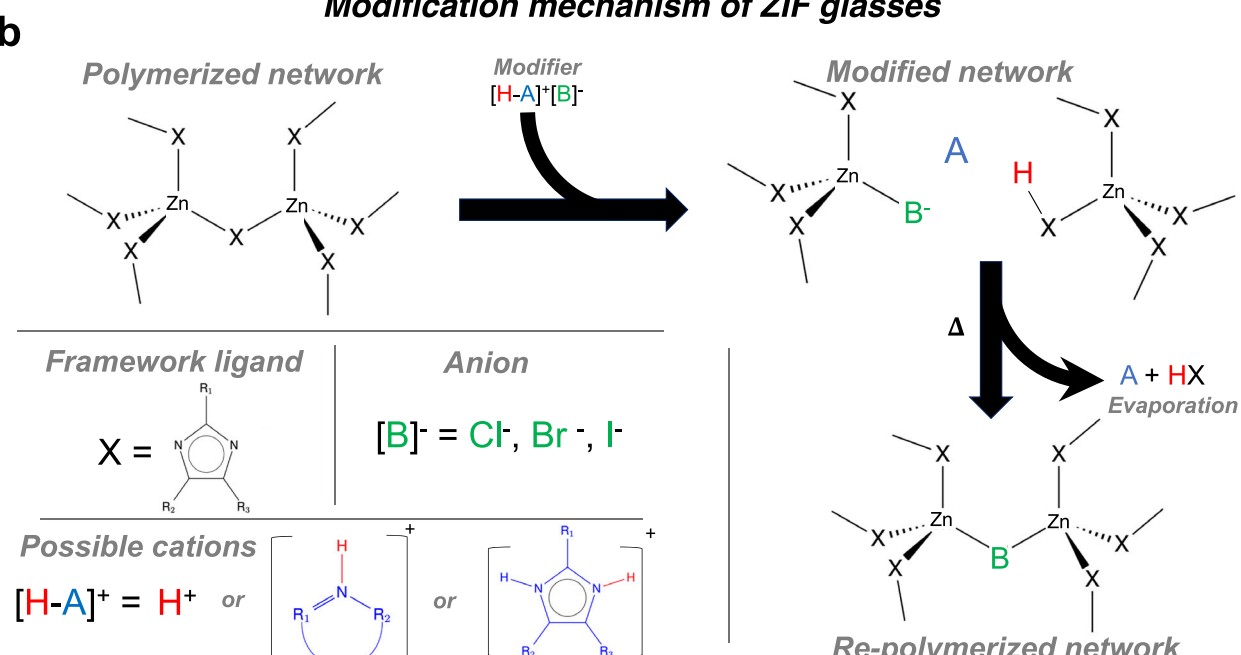

**Fig. 4 | Proposed mechanism of modification for ZIF glasses using halide salts.** **a** Established mechanism of modification of archetypical silicate glasses using alkali oxides. **b** Proposed halide salt modification mechanism of ZIF glasses. Addition of the halide salt partially breaks up the Zn-ligand-Zn bridges to form depolymerized structures, while further heating appears to induce the evaporation of the linker species and subsequent formation of halide-bridged Zn centers.

$R = 0.5$ system at 1000 K. The simulated system consisted of a single unit cell of crystalline ZIF-62 (296 atoms) mixed with 8 benzimidazolium species and 8 $Cl^-$ anions. After 10 ps of simulation, we find an overall decrease in energy (converged after ~6 ps, see Supplementary Fig. 40) and that all 8 $Cl^-$ atoms are coordinated to Zn atoms (bond length of ~2.2 Å), with one case of an imidazolate linker obtaining a hydrogen atom from the added benzimidazolium. Based on this simulation, we show the partial PDFs of the most relevant atomic pair-correlations in Fig. 3e. These results were also used in the assignment of the experimental peaks in Fig. 3b, c, confirming that the observed bond lengths correspond very well with newly formed peaks in the ZIF-62-bImCl ($R = 0.5$) glass. Lastly, based on the performed DFT-MD simulation, we present an example configuration of a Zn node coordinated with a mixture of (benz)imidazolate and chloride (Fig. 3f). Interestingly, in the DFT-MD simulation, we observe some cases of Zn-Cl-Zn bridges (like the bonding in crystalline $ZnCl_2$). We hypothesize that these may also be formed experimentally upon evaporation of free or non-bridging (benz)imidazole species (as observed experimentally, see Supplementary Text on Structural analysis by FTIR spectroscopy). This explains the observed increase in $T_g$ of the modified ZIF-derived glasses upon subsequent annealing (see Supplementary Fig. 7 and Supplementary Text), since the structure should become more rigid when non-bridging imidazolate linkers are replaced by bridging halide species.

### Mechanism of ZIF modification

Based on the experimental and simulation data, we propose the ZIF structural modification mechanism shown in Fig. 4 and include a comparison to the well-known modification of silicate glasses by alkali modifiers (Fig. 4a). The polymerized ZIF network consists of zinc metal nodes connected by bridging organic linkers (X). The addition of the modifier halide salt consisting of a proton-containing cation (here, a heterocycle denoted [H-A]$^+$), which may easily exchange its proton (H$^+$), together with a halide anion (denoted [B$^-$]), results in a partially depolymerized ZIF network. That is, upon heating, the breaking and reformation of Zn-N coordination bonds lead to changes in the coordination framework. During this process, Zn nodes coordinate to mixed linkers of protonated existing linkers (X-H), heterocycles (A, case not shown directly in Fig. 4) as well as the added halide anions (B). Continued heating of either the existing liquid or the already-formed glass (below the decomposition temperature) may result in evaporation of both the heterocycle (denoted as A) (as shown in Supplementary Fig. 41) as well as the X-H species, ultimately leading to re-polymerization of the framework. We note that the re-polymerization mechanism at high temperatures (lower right corner of Fig. 4b) needs further verification in future work. In any case, at high modifier concentration ($R = 1.0$), the structural framework undergoes a transition where Zn coordination is due to halide-anion linkages rather than imidazolate linkers. This shift in connectivity aligns with the depolymerization mechanism, where increasing halide content disrupts the extended ZIF network and promotes Zn-X coordination.

## Discussion

In future work, the herein proposed concept for ZIF glass modification may allow for a significant enhancement of the structural diversity of MOF glasses in general. Specifically, all the systems we have tested (ZIF-4, ZIF-8, and ZIF-62 co-melted with various heterocycle cations and halide [Cl, Br, I] anions) melt and form cm-sized homogeneous glasses. Given the equivalent possibility of, e.g., carboxylate-MOFs to form mixed carboxylate-halide-node metal centers[50], we believe that the present discovery should allow for glass formation in a broad range of MOF chemistries. This would allow MOF glasses to realize the hallmark of traditional glasses, namely, the ability to tune chemical and physical properties through continuous composition variation. We

also believe that the present approach will enable improved functionalities (fluorescence, sensing, etc.) of MOF glasses through the incorporation of the added salt directly into the existing framework structure.

In summary, we have demonstrated that the addition of an organic halide salt ("modifier") to three typical ZIF crystals results in meltable systems, which upon cooling form glasses with tunable thermal and mechanical properties depending on the composition and thermal treatment. Importantly, this approach enables the melting of ZIF-8, which otherwise undergoes decomposition before melting. The approach thus provides a different paradigm for structural tuning and processing of metal-organic frameworks.

## Methods

### Materials

All chemicals were used as received from the supplier without further purification. $Zn(NO_3)_2 \cdot 6H_2O$ (≥99.0%), methanol (99.9%), imidazole (99.5%), benzimidazole (98%), 2-methylimidazole (99.0%), HCl (hydrochloric acid, 37%), HBr (hydrobromic acid, 48%), HI (hydriodic acid, 57 wt% in $H_2O$, distilled, stabilized, 99.95%), and dimethylformamide (DMF) (99.9 %) were all acquired from Sigma-Aldrich.

### Synthesis of ZIF-62 crystal

Synthesis was performed according to the procedure described in Madsen et al.[51]. The samples were prepared by weighing zinc nitrate hexahydrate (1.7460 g), imidazole (5.3282 g), and benzimidazole (1.6268 g) and adding them sequentially to a beaker, followed by the addition of 50 mL of DMF. The solution was stirred for 30 min, then transferred to a 100 mL PTFE-lined autoclave and sealed. The mixture was placed in an autoclave at 130 °C for 96 h. Afterwards, the autoclave was allowed to cool naturally in the oven to ambient temperature overnight. The synthesized crystals were recovered and washed three times with ~40 mL of DMF each time. The centrifugation step was performed at ~7010 g. The washed crystals were dried in an oven at 110 °C overnight. X-ray diffraction results of the obtained material are presented in Fig. 1b.

### Synthesis of ZIF-4 crystal

The synthesis of ZIF-4 followed the method described in Widmer et al.[52]. 1.2 g of $Zn(NO_3)_2 \cdot 6H_2O$ (4.03 mmol) and 0.9 g of imidazole (13.2 mmol) were dissolved in 75 mL of DMF and transferred to a 100 mL PTFE-lined autoclave and sealed. The autoclave was sealed and heated in an oven at 130 °C for 48 h. The synthesized crystals were recovered and washed three times with ~40 mL of DMF. The centrifugation step was performed at ~7010 g. The washed crystals were dried in an oven at 110 °C overnight. X-ray diffraction results of the obtained material are presented in Fig. 1b.

### Synthesis of ZIF-8 crystal

ZIF-8 synthesis followed a procedure adapted from Venna et al.[53]. Initially, 0.3 g of $Zn(NO_3)_2 \cdot 6H_2O$ was dissolved in 11.3 g of methanol. Subsequently, 0.66 g of 2-methylimidazole and another 11.3 g of methanol were added to the zinc-based solution. After a reaction time of 12 h, the gel formed in the solution was centrifuged at ~7010 g to separate the crystals. The crystals were then washed three times with methanol. The washed crystals were dried at 75 °C overnight. X-ray diffraction results of the obtained material are presented in Fig. 1b.

### Synthesis of benzimidazolium chloride salt

The synthesis of benzimidazolium chloride ($H_2bImCl$) followed the method described by Peppel et al.[54]. Benzimidazole (8.671 g, 73.4 mmol) was dissolved in 25 mL demineralized water in a beaker, followed by the addition of 37% hydrochloric acid (7.95 g, 81.2 mmol) with stirring. The solutions were heated to 110 °C in an oil bath until all liquid had evaporated, resulting in the formation of white crystalline

powder. The powder was dried in a vacuum environment at 110 °C for 24 h before being subjected to X-ray diffraction analysis as presented in Supplementary Fig. 42.

## Synthesis of benzimidazolium bromide salt

The synthesis follows the same strategy of an acid-base salt as for benzimidazolium chloride to obtain benzimidazolium bromide ($H_2bImBr$). Benzimidazole (8.671 g, 73.4 mmol) was dissolved in 25 mL demineralized water in a beaker, followed by the addition of 48% hydrobromic acid (13.686 g, 81.2 mmol) with stirring. The solutions were heated to 110 °C in an oil bath until all liquid had evaporated, resulting in the formation of brown crystalline powder. The powder was dried in a vacuum environment at 110 °C for 24 h before being subjected to X-ray diffraction analysis as presented in Supplementary Fig. 42.

## Synthesis of benzimidazolium iodide salt

The synthesis follows the same strategy of an acid-base salt as for benzimidazolium chloride to obtain benzimidazolium iodide ($H_2bImI$). Benzimidazole (8.671 g, 73.4 mmol) was dissolved in 25 mL demineralized water in a beaker, followed by the addition of 57% hydriodic acid (18.222 g, 81.2 mmol) with stirring. The solutions were heated to 110 °C in an oil bath until all liquid had evaporated, resulting in the formation of gray-black crystalline powder. The powder was dried in a vacuum environment at 110 °C for 24 h before being subjected to X-ray diffraction analysis as presented in Supplementary Fig. 42.

## Preparation of modified ZIF-derived glasses

The utilized heating source was a hotplate (IKA C-MAG HS7 hotplate stirrer), upon which an aluminum block with drilled holes was positioned to ensure uniform heating of the test tube (see Supplementary Fig. 43 for an illustration of the setup). Approximately 0.2 g of ZIF crystal and varying amounts of halide salt (depending on the desired molar ratio $R$) were accurately weighed and then ground using a mortar and pestle to achieve a homogeneous mixture. The mixture was subsequently transferred into the test tube, which was sealed with a soft plug. Argon gas was purged into the test tube to remove any air, and then the test tube was placed in the preheated aluminum block at a predetermined temperature for heating and melting of the mixture. Finally, the test tube was cooled to room temperature to obtain the hybrid glass by simply removing the tube from the aluminum block.

## Fast scanning differential calorimetry

All fast scanning differential calorimetry (FDSC) measurements were done using a Mettler Toledo Flash DSC 2+ device. Samples were initially crushed under a scalpel blade. For each sample, a single grain (microgram range) of the homogenous sample was loaded onto the active area of a Mettler Toledo pre-conditioned and pre-corrected UFS1 sensor using a hair (see example of a loaded sample in Supplementary Fig. 44). The sample was then prepared at temperatures ranging from 200 to 300 °C i.e., well above $T_g$ of the modified ZIF-derived glasses, to establish a good thermal contact between the sensor and the sample. All sample scans were performed in a nitrogen atmosphere (flow of 20 mL min$^{-1}$). The presented FDSC results are shown uncorrected. Glass transition temperatures are reported as onset temperatures, i.e., the intersection between the extrapolated lines of the heat flow in the glassy state (significantly below $T_g$) and the tangent of the inflection point of the glass transition peak. For clarity, the different FDSC scans of the same sample are plotted using the recorded heat flow (typically in the range of 0.1–2 mW depending on sample size and heating rate) but plots with data for different samples are normalized by the maximum heat flow of each scan. This allows for better comparison of the samples, given the inability to normalize the

heat flow data by sample mass (due to the extremely small sample size used in FDSC).

## Standard differential scanning calorimetry

Standard DSC measurements were performed using a Netzsch STA F449 F3 instrument equipped with liquid $N_2$ cooling. Measurements were performed in Netzsch cold-welded aluminum crucibles in a He atmosphere except for measurements conducted to obtain isobaric heat capacity, which used PtRh crucibles. Heating and cooling rates of 10 K min$^{-1}$ were used. The thermal history of the measured glass was "reset" by heating to 200 °C (i.e., >>$T_g$) before cooling at a rate of 10 K min$^{-1}$ and finally acquiring a heating scan at 10 K min$^{-1}$. Measurements of absolute heat capacities were obtained using sapphire as a standard material.

## X-ray diffraction

X-ray diffraction (XRD) measurements of the samples were performed on a PANalytical Empyrean X-ray diffractometer with Cu Kα ($\lambda = 1.5406$ Å) radiation. The XRD patterns were collected in the $2\theta$ range of 5–40° with a step size of 0.013°.

## X-ray total scattering

X-ray total scattering measurements were conducted at the P02.1 beamline at PETRA III, Deutsches Elektronen-Synchrotron (DESY) in Hamburg, Germany. The data were collected on an amorphous silicon two-dimensional flat panel Varex XRD 4343CT (2880 × 2880 pixel matrix for 150 × 150 μm$^2$ pixel size) in corner configuration with a quarter ring $Q_{max}$ of 28.1 Å$^{-1}$. The beam spot was 1 × 1 mm$^2$ and the used wavelength $\lambda$ was 0.207 Å (60 keV). In situ temperature-resolved measurements were performed in quartz glass capillaries (QGCT 1.0, Capillary Tube Supplies Ltd) with an inner diameter of 0.9 mm and a 0.1 mm wall thickness, whereas the glass samples measured ex situ at room temperature were placed in Kapton capillaries (Cole-Parmer Instrument Company, Polyimide tubing 1.2 mm) with an inner diameter of 1.03 mm and a 0.051 mm wall thickness to achieve good transmission of X-rays. The in situ measurements were performed using an exposure time of 60 s at each 10 K temperature step. Heating and cooling between steps were performed at a ramp of 25 K min$^{-1}$ using a hot air blower[55]. Real and setpoint temperatures were checked using the (2 0 0) Bragg reflection of a copper sample. We found a difference between real and setpoints temperatures of <10 K (Supplementary Figs. 45, 46) for all measurements and we thus used the setpoint temperature for all reported temperatures of the total scattering data. The ex situ measurements were performed at room temperature using an exposure time of 600 s.

To process the obtained data, azimuthal integration and calibration were made using pyFAI[56] software. LaB$_6$ was used as a calibrant for the detector to sample distance and angle. xPDFsuite[57] was used for subtraction of the empty quartz glass and Kapton capillaries integration intensities and to obtain the $S(Q)$ ($R_{poly} = 1.20$). The Fourier transform was made over the $Q$-range of 0.1 to 20 Å$^{-1}$ of the reduced total scattering structure function ($F(Q) = Q[S(Q) - 1]$) as obtained from xPDFsuite. This procedure was performed using an in-house code. A Lorch window function[58] (with $Q_{max} = 20$ Å$^{-1}$) was used to process the ex situ data (Fig. 3c, d) to reduce the effect of termination ripples in the resulting $G(r)$ from the Fourier transform. All in situ data (Fig. 3b and all PDFs in the Supplementary Information) are presented without the use of a Lorch function. We note that a few of the samples exhibited a small extent of crystallization (ZIF-62-bImCl $R = 1.0$ and ZIF-62-bImBr $R = 1.0$), which is likely caused by the prolonged heating in the beamline setup compared to preparation on the hot plate or even in the standard rate DSC. Despite this minor crystallization, we find an absence of long-range order in the PDFs up to higher pair correlation distances (30 Å, see Supplementary Fig. 47 for both samples) when compared to the ingoing mixtures.

## X-ray photoelectron spectroscopy

Before each X-ray photoelectron spectroscopy (XPS) measurement, the glass samples were polished using SiC paper with anhydrous diamond suspension and hexane to obtain a smooth surface. The crystalline samples were dried and ground into powder using a mortar. The XPS measurements were performed using a Hiden MAXIM SIMS system equipped with a Specs XR50 X-ray source and a Specs Phoibos 150 electronic analyzer. The X-ray beam generated by the aluminum anode had a wavelength of $\lambda = 0.83401$ nm, and X-ray optics ensured signal collection from spots ~2 mm in diameter, including glass samples. Subsequent data analysis was performed using CasaXPS software, involving standard energy correction and background removal procedures based on the expected C1s transition (284.8 eV).

## Raman spectroscopy

Micro-Raman spectra were recorded with a diode laser of 532 nm wavelength equipped on a Renishaw Invia spectroscope.

## Fourier transform infrared spectroscopy

FTIR spectra were recorded using an attenuated total reflection setup on a Bruker Tensor II spectrometer. Crystalline diamond was used as the attenuation crystal. All samples were measured under ambient conditions in the 400–4000 $cm^{-1}$ frequency range.

## Far infrared spectroscopy

FIR spectra were recorded using a Thermo Fisher Scientific Nicolet iS50 instrument. All samples were measured under ambient conditions in the 100-450 $cm^{-1}$ frequency range.

## Solution $^1$H NMR

Solution $^1$H NMR spectra of digested samples [in a mixture of DCl (35%)/$D_2O$ (0.1 ml) and dimethyl sulfoxide-$d_6$ (DMSO-$d_6$; 0.5 ml)] were performed. In detail, spectra of desolvated crystalline samples and the glasses (about 10 mg each) were recorded on a Bruker Avance III 600 MHz spectrometer at 308 K. Chemical shifts were referenced to the residual solvent signals of non-deuterated DMSO. The spectra were processed with the MestreNova Suite.

## Solid-state $^1$H MAS NMR

Solid-state $^1$H MAS NMR analyses were carried out on a 22.3 T Bruker Avance NEO (950 MHz for $^1$H) spectrometer equipped with 1.9 mm HCND four-channel probe. Direct pulse MAS (35 kHz spinning frequency) experiments (single 90-degree pulse; pulse length/amplitude 4 μs/50 kHz; 3 s repetition delay; 16 scans) were used for acquiring the experimental data. The isotropic chemical shifts are relative to $^1$H signal for adamantane (1.82 ppm).

## Solid-state $^{13}$C MAS NMR

Solid-state $^{13}$C MAS NMR analyses were carried out on a 16.4 T Bruker Avance III HD wide-bore (700 MHz for $^1$H) spectrometer equipped with 4 mm HXY triple-resonance probe in double-resonance mode at MAS frequency of 15 kHz. $^1$H-$^{13}$C Cross-Polarization was used for acquiring experimental data using a field strength of 64 kHz on $^1$H and 50 kHz on $^{13}$C with an 80–100% RAMP on $^1$H channel (contact period of 0.75 ms). $^1$H decoupling (SPINAL64, 110 kHz) was employed during the free acquisition time. Spectra were recorded using a repetition delay of 3 s. A total number of 312 scans (ZIF-62 glass) and 912 scans (ZIF-62-bImCl $R = 1.0$ glass) were acquired. The isotropic chemical shifts are relative to $^{13}$C signal for adamantane (37.8 ppm).

## Solid-state $^{15}$N MAS NMR

Solid-state $^{15}$N MAS NMR analyses were carried out on a 16.4T Bruker Avance III HD wide-bore (700 MHz for $^1$H) spectrometer equipped with 4 mm HXY triple-resonance probe in double-resonance mode at MAS frequency of 15 kHz. $^1$H-$^{15}$N Cross-Polarization was used for acquiring

experimental data using a field strength of 35 kHz on $^1$H (with 80–100% RAMP) and 18 kHz on $^{15}$N channel (contact time of 10 ms). $^1$H decoupling (SPINAL64, 110 kHz) was employed during the free acquisition time. ZIF-62 glass spectra used a repetition delay of 2 s and were recorded using 4096 scans. For the ZIF-62-bImCl $R = 1.0$ glass, spectra with a repetition delay of 5 s and 45,769 scans were acquired. The isotropic chemical shifts are relative to a solid sample of $NH_4Cl$ set to 39.3 ppm.

## Solid-state $^{67}$Zn MAS NMR

Solid-state $^{67}$Zn MAS NMR analyses were carried out on a 22.3 T Bruker Avance NEO (950 MHz for $^1$H) spectrometer equipped with 4 mm HX double-resonance probe at MAS frequency of 15 kHz. The data were recorded using a Hahn-echo pulse sequence using 17 kHz rf field strength with a total echo-time of one rotor period (lengths of 2 and 4 μs for the $\pi/2$ and $\pi$ pulses, respectively). The spectra used a repetition delay of 0.1 s. In addition, an empty rotor experiment was recorded and subtracted. The isotropic chemical shifts are relative to an aqueous 1.0 M solution of $Zn(NO_3)_2$.

## Scanning electron microscopy and energy dispersive X-ray analysis

For investigations of the surface morphology and elemental composition of the samples, SEM and EDX measurements were conducted. The samples were coated with 10 nm platinum to improve the electrical conductivity. A Tescan Clara UHR SEM was used for the acquisition of SEM images and EDX spectra. The instrument was operated in analysis mode at a voltage of 20 keV and a beam current of 300 pA. A working distance of around 10 mm was used. SEM images were acquired with an Everhart-Thornley type (YAG Crystal) detector. The EDX spectra for the compositional maps were acquired with an Oxford Ultim Max 40 mm$^2$ EDS detector and analyzed with a AztecLive Standard software.

## Micro indentation

Vickers hardness ($H_V$) and crack initiation resistance (CR) of the samples were measured by micro-indentation (CB500, Nanovea). For hardness tests, we used a Vickers diamond tip (four-sided pyramid-shaped diamond with an angle of 136° between opposing faces) to produce indents at varying peak loads on each sample, with a loading rate of 0.05 N min$^{-1}$ and a holding time of 6 s. 15 indents were analyzed for each sample. The lengths of the two indent diagonals were measured using an optical microscope. $H_V$ (in GPa) was then calculated as,

$$H_V = \frac{0.1891P}{d^2},$$

where $P$ is the used indentation force (in N) and $d$ is the average indent diagonal length (in mm). The CR value corresponds to the load value with a probability of corner cracking of 50%[59]. Specifically, the initiation probability of crack initiation is defined as the ratio between the number of cracked corners and the total number of corners during the test. The measurement of CR was also done using the Vickers tip, but here we used varying peak loads. The number of corners with cracks after unloading at different loads and different loading rates was observed using a microscope. The indentations were performed under laboratory conditions (23 °C, ~40% relative humidity). The ratio between the number of cracked corners and the total number of corners was recorded for each load 1 h after unloading. For each sample and load, 15 indents were analyzed.

## $CO_2$ adsorption

$CO_2$ adsorption isotherms were collected at 231 K (−42 °C), 273 K (0 °C), and 295 K (22 °C, room temperature) using an Autosorb iQ2 Automated Gas Sorption Analyzer (Quantachrome Instruments). Prior

to measurements, all samples underwent 168 hours of degassing under dynamic vacuum at room temperature to eliminate residual adsorbates. Full adsorption–desorption isotherms were collected with 80 data points per isotherm, ensuring high-resolution adsorption profiles across the relative pressure range $P/P_0 = 0.01–1.0$. To express $CO_2$ adsorption in $mmol\,g^{-1}$, the volume ($cm^3\,g^{-1}$) at standard temperature and pressure values were converted using the ideal gas law.

## DFT-based molecular dynamics simulations

We performed DFT-based molecular dynamics (MD) simulations of a unit cell of ZIF-62, including eight structures of added benzimidazolium chloride. The initial structure was made using packmol[60], where benzimidazolium and chloride were placed randomly in a crystalline unit cell of ZIF-62 while keeping a minimum distance of 2 Å to the existing atoms. The dynamics simulations were then performed with the Vienna Ab initio Simulation Package (VASP) using the standard PBE pseudopotential and a time step of 0.5 fs. An energy cutoff of 400 eV and a convergence criterion of $10^{-4}$ eV were used. First, the structure was relaxed while allowing relaxation of both unit cell dimensions and positions using a target pressure of 0. Next, dynamics were initiated at 1000 K and run for 1 ps to equilibrate the structure before running 10 ps (a total of 20,000 molecular dynamics steps) of dynamics at 1000 K to probe the structural changes between the ZIF and the benzimidazolium chloride modifier. In this process, the structure was allowed to deform freely with a target pressure of 0.

## Data availability

The data generated in this study are provided in the Main text and Supplementary Information. During the present study, we used the following CIF files to simulate X-ray diffraction patterns for experimental comparison as obtained from the CCDC database: CCDC Deposition number 602538 (ZIF-4 CIF), CCDC Deposition number 1429243 (ZIF-8 CIF), CCDC Deposition number 671070 (ZIF-62 CIF). Source data are provided with this paper.

## Code availability

The code used in the performed VASP calculations is available at https://github.com/OxideGlassGroupAAU/Modified-ZIF-simulation under https://doi.org/10.5281/zenodo.15707617.

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

## Acknowledgements

We acknowledge M. Umair and L. Piemontese (University of Padova) for assistance with calorimetry measurements, N. Bjerre-Christensen and D. Ravnsbæk (Aarhus University) for preliminary X-ray total scattering measurements, and Z. Li (Aalborg University) for assistance with liquid NMR measurements. M.M.S. acknowledges support from the European Union (ERC, NewGLASS, 101044664), the ESS lighthouse on hard materials in 3D, SOLID, funded by the Danish Agency for Science and Higher Education (8144-00002), and Independent Research Fund Denmark (1127-00003). Views and opinions expressed are however, those of the author(s) only and do not necessarily reflect those of the European Union or the European Research Council. Neither the European Union nor the granting authority can be held responsible for them. F.C. acknowledges support from the China Scholarship Council (202206890034). N.C.N. and A.B.N. acknowledge support from the Novo Nordisk Foundation NERD program (NNF22OC0076002) and the use of NMR facilities at the Danish Center for Ultrahigh-Field NMR Spectroscopy funded by the Danish Ministry of Higher Education and Science (AU-2010-612-181) and Novo Nordic Foundation Research Infrastructure— Large Equipment and Facilities (NNF22OC0075797). S.S.S., F.C., and M.M.S. acknowledge DESY (Hamburg, Germany), a member of the Helmholtz Association HGF, for the provision of experimental facilities and travel support. Parts of this research were carried out at PETRA III beamline P02.1. Beamtime was allocated for proposal I-20230857 EC. A.K.R.C., S.S.S., F.C., S.M., M.A.K., V.B., and X.G. acknowledge the Danish Agency for Science, Technology, and Innovation for travel funding through DanScatt. S.S.S. and M.M.S. acknowledge the computational resources supplied by Aalborg University (through CLAAUDIA) and access to the LUMI HPC through DeiC (grant no. DeiC-AAU-N5-000006) as well as funding from the Carlsberg Foundation (CF21-0371) for the standard DSC instrument. R.K. acknowledges the Carlsberg Foundation (CF20–0364) and Aarhus University Centre for Integrated Materials Research (iMat) for the Tescan Clara SEM instrument.

## Author contributions

S.S.S. conceived the idea of the project. F.C., S.S.S., and M.M.S. planned the study. F.C. performed all sample synthesis, DSC, XRD, FTIR, far-IR, and Liquid NMR measurements and analyses. S.S.S. performed atomistic simulations. R.K. and D.S. performed SEM and EDX measurements. N.L. and R.L performed gas adsorption experiments. S.S.S., F.D., J.B., G.M., and F.C. performed FDSC measurements and analyses. P.K.K. performed XPS measurements and analyses. L.R.J. and F.C. performed Raman spectroscopy measurements and analyses. A.B.N. and N.C.N. performed solid-state NMR measurements and analyses. A.K.R.C., S.S.S., F.C., S.M., M.A.K., V.B., and X.G. performed X-ray synchrotron measurements and analyses. F.C. and D.S. performed the micro-indentation experiments and analyses. X.G. participated in part of the experimental design. S.S.S. and M.M.S. supervised the study. F.C.,

S.S.S., and M.M.S. wrote the manuscript, with input from all other authors. All authors discussed the results.

## Competing interests

The authors declare no competing interests.
