## [Transparent Peer Review file · Nature Communications]

Continuous structure modification of metal-organic framework glasses via halide salts

Corresponding Author: Professor Morten Smedskjaer

Version 0:

Reviewer comments:

Reviewer #1

(Remarks to the Author)

This is a nice paper of potentially broad interest across the field of glass science, materials engineering and beyond. The authors adopt empirical tools of classical glass chemistry in order to extend the range of MOF glass compositions – MOF glasses have emerged as a new class of glasses potentially combining the properties of metal-organic framework materials with the processing ability of glasses, but the range of MOF chemistries from which glasses can actually be produced has been limited to an extent which is currently preventing materials design through chemical formulation.

I have one sole concern with the paper, of which I am sure the authors can provide appropriate answers: in order to truly extend the range of available MOF glasses, the authors certainly agree that it is not sufficient to produce further glass-forming compounds which include a MOF as an ingredient, but which do not retain structural and physical properties of this MOF, in particular, its porosity. That said, a MOF glass is probably strictly only a MOF glass when it exhibits a metal-organic framework backbone.

According to the author's approach – with similarity to the classical network hypothesis developed for oxide glasses, in which the glass network can be "modified" using "network modifying species" – adding halide compounds to the MOF glass would reduce the network polymerization grade in similar ways as found in oxide glasses. In oxide glasses, the classification of what makes a network modifier is mostly empirical, being based on a lower threshold of oxide bond strength. This makes the current approach much more clear in that there is a clear-cut difference between the (MOF) network and the salt modifier. That said, the 100% modified material in Fig. 1d should surely not be termed "ZIF", and it remains to be shown that the other two data points (50% and 25% modified) indeed retain a ZIF network topology. I am not convinced that the NMR and PDF analyses presented in Fig. 3 provide such proof, in particular, as they are integral measurements from which it is not immediately clear whether the observed mid/longer-range features originate from a homogeneous material or from a (nano-)composite. But this puts into question the scheme proposed in Fig. 4. The same holds for the further short-range structural data provided in the SI.

As a pragmatic solution, I would love to see physical property data, for example, glass adsorption, permeation and/or porosity analyses from which retained MOF porosity could be confirmed. The sample photographs indicate relatively good material quality; I suggest that optical scattering spectra are provided on these samples in order to evaluate material homogeneity (vs. a potential composite nature) – Fig. S7 is perhaps a good start.

The R-parameter is not well defined. Perhaps it can be related to charge balance / charge compensation, or the Q-group speciation similar to the way this is done in oxide glass chemistry?

I applaud the authors for the quality of their flash-DSC data! However, beyond Fig. S39 there is only little information on low-rate DSC, and there is no heat capacity data. If I understood correctly, all data provided are normalized; would it be possible to provide the non-normalized data, too?

A minor point: I do not agree that the MD simulations should be called "ab initio", even when this is often done elsewhere.

Reviewer #2

(Remarks to the Author)

The manuscript by Sørensen, Smedskjaer and co-workers reports a new protocol to modify the melting of crystalline ZIFs by incorporation of additives, in a similar way to what is currently performed in oxide-based glasses. In order to prove this approach in ZIFs, the methodology has been generalized by using three different ZIFs (namely ZIF-4, ZIF-8 and ZIF-62), three different additives (benzimidazolium chloride, benzimidazolium bromide and benzimidazolium iodide) and different

compositions ZIF:additive. This study is performed using Fast Differential Scanning Calorimetry (FDSC) in order to achieve the melting of the mixtures in a very short time, instead of using the common Differential Scanning Calorimetry (DSC) that has been typically used to report the melting of the ZIFs.

The idea of the work is interesting and could provide a way to modify the physical properties of this attractive type of materials. However, the use of the non-standard FDSC to produce the glassy form of the materials is not very convenient, as the results shown by the authors could be biased by this. Furthermore, as will be detailed below, the characterization of the materials does not show unequivocally the formation of the glass that the authors are proposing.

1. The authors justify the choice of FDSC with two reasons (described in the Supplementary Text): the presence of weak glass transitions and the limitation of permanent structural changes. If a new methodology, as the proposed in this manuscript, is presented, standard systems should be used, not those with weak glass transitions. If the glass transitions change from clearly observable to weak, that could mean that the methodology is not suitable for this type of materials. In addition, the defense that FDSC is useful to limit permanent structural changes is again non-justifiable, as the new methodology should be valid in standard conditions (this should not be restricted to a heat speed of 500 K/s; if that is the case, then this work would not be of general interest).
2. The characterization of T_g of the different samples has been performed using only one FDSC cycle. The T_g of some of these materials is rather unusual (e.g. 60 °C) and thus should be confirmed with additional DSC cycles. Again, the use of standard DSC instead of FDSC could facilitate this. In fact, the authors state, "We note how this melting temperature is below that of most other known ZIF systems," and the result is compared with traditional ZIFs. However, it is misleading to draw conclusions from such a comparison, as FDSC is not directly comparable with conventional DSC.
3. A fundamental technique to characterize the stability of the materials is Thermogravimetric Analysis (TGA), which provides information on the decomposition temperature. Are the materials stable enough to be heated at 500 K/s without any decomposition? Could the signal observed at 60°C be related with a partial decomposition of the ligands? In this sense, the black colour of the material resulting from heating ZIF-8 and the additives (e.g. Figure 2a, Supp. Fig. 2, or Supp. Fig. 10) suggests carbonization of the organic ligands. Thus, the nature of this solid is unclear.
4. Solid state NMR has been used for the characterization of the solids, but the information that this technique provides is quite limited. In this case, the authors should provide liquid-state NMR of the digested samples to clarify the ratio of imidazole (or methylimidazole) and benzimidazole incorporated in the final material for the different compositions.
5. The presence of Zn-Cl, Zn-Br, and Zn-I peaks in the PDF is quite revealing (Figure 3d) and provides strong evidence for the incorporation of halogens into the framework. However, EDX analysis should also be performed, in order to quantify the Zn:halide ratio of the final material. Is there a "ZnX₂N₂" unit, or is it a "ZnXN₃" unit? Or are there mixtures of all possible compositions? In addition, the electronic images are somewhat unclear, and it is difficult to visualize the distribution of atoms (Figure 2).
6. It is unclear the final type of material that is formed in the glass. The additive that is used is composed of benzimidazole and halogen units. The halogen units are incorporated to the material, partly bound to Zn (although as stated in point 5, it is not clear how many halogens are bonded to Zn), and maybe part is uncoordinated (not revealed). What about the benzimidazole part? They could be bonded to Zn, or not.
7. In Figure 1, the additive is described as "benzimidazolite halide" but this should be "benzimidazolium halide". Also, the short version "benzimidazolium halide" could be clearer if H₂blmCl was used instead of blmCl.

Reviewer #3

(Remarks to the Author)

In this work, Cao et al. present a thorough and insightful study on the continuous structural modification of metal-organic framework (MOF) glasses via the introduction of halide salts as chemical modifiers. Through a combination of variable temperature X-ray total scattering and NMR data, the authors effectively demonstrate the coordination of the modifier with Zn^{2+} . This innovative method may broaden the landscape of MOF glass chemistry and may offer a platform for tuning the thermal and mechanical properties of these materials. However, some concerns remain regarding the thermal and structural analysis. In this regard, the manuscript may be considered for publication in *Nature Communications* after major revisions. The authors should consider addressing the following points to strengthen their work:

1. In page 3, line 67, It would be beneficial to clarify that the pure phase of ZIF-76 cannot be melted (*CrystEngComm*, **2020**, 22, 3627-3637; *Angew. Chem. Int. Ed.* **2024**, 63, e202405307). In the original study, the melting observed was for a non-pure phase of ZIF-76 (*Nat. Commun.*, **2018**, 9, 5042). Additionally, the authors seem to ignore the fact that ZIF-8 is meltable (*Nat. Commun.*, **2024**, 15, 4420) after linkers exchange in their subsequent presentation of the effect of linker ratios on melting and the melting of ILs@ZIF-8.
2. It is noted that the glasses were prepared using a hotplate, which precluded the authors from capturing DSC signals during the first heating of the mixture. It would be good to measure the heat flow during the first heating using a closed aluminium crucible or high-pressure crucible to determine whether the modifiers melt individually or if there is a single co-melting endothermic signal. Additionally, could there be thermal behavior signal from the coordination of Zn^{2+} with halide ions? In other word, when would the reaction ZIF with halide salts occur, at the room-temperature ground preparation or the heating procedure of the mixture? The current understanding of the co-melting process, based only on variable temperature X-ray total scattering with full amorphisation around 300°C , is insufficient. Moreover, in Figure 1c, would it not be more appropriate to refer to a "supercooled liquid" rather than a "liquid"?
3. We also suggest measuring the change in heat capacity (ΔC_p), as this would provide insight into the degree of connectivity within the MOF glasses. How do the ΔC_p values of the modified ZIF glasses compare to those of pure ZIF-62 glass? It remains unclear whether this experiment can be conducted using FDSC, although the authors note that FDSC's fast heating rate allows the detection of weaker heat flow signals. Furthermore, how to define and/or differ the vitrification from melting for these ZIF-bImCl R=1 cases in this study? Additional discussion of which may be beneficial and more clear for common readers.
4. In Fig. 3b, where VT-PDF data for ZIF-62-bImCl R=0.5 is presented, there are eight temperature points shown as $S(Q)$ in Supplementary Fig. 18, yet only seven $G(r)$ data points are displayed in Fig. 3b. Additionally, the decision to switch from R=1.0 to R=0.5 in the main text appears to avoid the fact that R=1 is not fully amorphous (Supplementary Figure 24). This phenomenon is similarly observed in ZIF-62-bImBr R=1 (Supplementary Figure 28). In light of this, we recommend displaying $G(r)$ data up to 30 or 50 Å to better assess the presence of long-range ordering. In the VT-PDFs, the Zn-Zn peaks at ~ 6 Å, often characteristic of ZIF glasses, are weak and asymmetric, raise doubts as to whether the final products are indeed ZIF glasses or coordination polymer (CP) glasses.
5. Page 12, line 283: The authors introduce 'imidazolium chloride' to examine the universality of their approach, yet the results appear divergent. In Supplementary Figure 31, PXRD data for both ZIF-8 and ZIF-4 reveal weak peaks around 15° , possibly attributable to the **zni** phase (ZIF-61 or ZIF-zni), for which no explanation is provided. Furthermore, no corresponding variable temperature X-ray total scattering data for ZIF-8-ImCl R=1 and ZIF-4-ImCl R=1 is offered. Also, the absent PXRD diffractions of ZIF-8-ImCl, ZIF-62-ImCl, ZIF-4-ImCl (R=1) before heating in Fig 1b, as well as supplementary Figure 1, make their unclear morphology or structure phase at room temperature. Are they glassy, gel or monolith?
6. In Supplementary Figure 11, the peak at 421 cm^{-1} is unlikely to be due to Zn-N stretching. Zn-N stretching typically occurs in the far-infrared region around $\sim 300\text{ cm}^{-1}$ (*Phys. Rev. Lett.* **2014**, 113,

- 215502). Far-infrared measurements are recommended to confirm the presence of Zn-N tetrahedra in the final glass. If absent, this would suggest the formation of a CP glass rather than a ZIF glass.
7. Is the dark colour of the ZIF-8 glass due to the decomposition of the methyl group? Solution ^1H NMR of the digested samples is recommended for all samples, as this would provide valuable information on both the integrity of the linkers and the exact composition of the samples.
 8. FT-IR spectra (Supplementary Figure 11b) of ZIF-8-bImCl R =1 are markedly weakened and different as compared to others, which should be addressed and revised in corresponding main text, or be rerecorded. Additionally, we noted the binding energy for C 1s of XPS data (Supplementary Figure 11b) significantly shifted.
 9. Confused label/names of glasses appeared in supplementary Figure 1, such as R=1 and R=1:1 shown, are they same as those in main text? Also, 'Tg of > 40K' in caption of Supplementary Figure 4 is not consistent with that shown in that plots.
 10. Through the author's experiments, it would be possible to remove additional modifiers from the glasses (*Nat. Commun.* **2024**, 15, 2040). Given the importance of porosity in ZIF glasses, it would be beneficial to provide CO₂ adsorption data at 195 K and the porosity information of the before/after modifiers removal.

Chong Ding Wan

Reviewer #5

(Remarks to the Author)

Version 1:

Reviewer comments:

Reviewer #1

(Remarks to the Author)

This has become a very nice manuscript. I greatly appreciate the way the authors handled my own and the other reviewer's comments and suggestions.

Reviewer #2

(Remarks to the Author)

The revised manuscript by Sørensen, Smedskjaer and co-workers has significantly improved with the new characterization techniques performed by the authors, which have served to clarify many of the points that were raised by the reviewers. Specially, one common point from all the reviewers was the lack of "standard DSC" measurements, which are now incorporated, and was one of the major criticisms.

However, the revised manuscript has still some issues that are not fully clear.

1. The composition of the glasses is not clear.

From the SEM-EDAX data shown in Supplementary Figure 9 (Response Figure R2), the Zn:Cl ratio is 2:1 for R=1, but this ratio should be 1, as $R = 1$ indicates that the number of moles of modifier and the number of moles of ZIF are the same (and there is 1 equivalent of Cl per modifier, and 1 equivalent of Zn per ZIF).

The proposed formula is also wrong. In the response 5 to reviewer 2, where a detailed description of the composition of the

different glasses is discussed, the formula used for this discussion is wrong: it is stated that the formula is $Zn(im)1.75(bim)1.25Cl$, but this formula indicates 4 negative charges and only 2 positive charges. This could be the result of part of the imidazoles/benzimidazoles being protonated and non-coordinated, e.g. something like $Zn(im)1.75(bim)0.25(H2bimCl)$, which is charge-balanced.

The authors also mention that there is uncoordinated benzimidazole in the final material. However, it is not clear if this has been taken into account in the liquid NMR of digested samples. In addition, the new TGA data (Supplementary Figure 36, Response Figure R8) shows a mass loss of 20% in the first scan. Could this be related with the uncoordinated ligands? Could this be the cause of the black colour observed in some samples?

2. New optical transmission data has been incorporated (Supplementary Figure 10, Response Figure R3), but the ZIF-8 derived glass has not been included here. This should be explained. Is it because of the black colour?

3. The new TGA data is not clearly presented: In Supplementary Figure 1 (Response Figure R5) this data is shown as dashed lines in a very small scale (right axis) which makes it extremely difficult to analyse. It seems there is some mass loss, but it's very difficult to tell. Also, in panels c and d of this figure, the data has a discontinuity at ca. 125 °C, which seems to be caused by the different scans in the DSC, but the values of the TGA do not seem to correspond to the end points of the first scans.

4. New adsorption isotherms (N₂ and CO₂) have been performed to study the porosity of the materials. The CO₂ sorption is presented for different materials in Supplementary Figure 14 (Response Figure R4), but the data is presented with different scales for the different solids. The unmodified glasses (panels c and d) show an uptake of 1.6 and 1.4 mmol CO₂/g, whereas the modified glasses novel in this study show an uptake of 0.03 mmol CO₂/g (panel e), 0.4 (panel f), 0.03 (panel g), 0.13 (panel h) and 0.04 (panel i), much lower than the original materials. In other words, the porosity is being lost. This figure should also be corrected, as Response Figure R4 has panels h and i which are missing in Supplementary Figure 14.

5. The compositional mapping from the SEM images should present a clear view of the solid with its edges, so it can be clearly observed the presence of the elements in the material and the absence of the elements where no material is present. The current images shown in Figure 2 (and in Response Figure R1) make it impossible to unambiguously determine the presence or absence of the elements.

6. The temperature dependent studies are shown in the different figures with the temperature indicated in the right hand side of the figure, in a graph. This is a nice way of presenting the data, but the authors might consider aligning the temperature points with the corresponding graphs of the left part, so it would be much easier the identification of the temperature (in addition to the different colour used)

Reviewer #3

(Remarks to the Author)

Reviewer #4

(Remarks to the Author)

The authors have made a commendable effort in revising the manuscript. They have conducted and included a substantial amount of additional experimental work, which has addressed nearly all of the concerns. The new data significantly strengthen the conclusions of the study and enhance its overall scientific value.

However, Figure 3b seems to have forgotten to be updated to 8 Gr data. With this minor correction, we believe the manuscript meets the high standards of Nature Communications.

Reviewer #5

(Remarks to the Author)

Version 2:

Reviewer comments:

Reviewer #2

(Remarks to the Author)

The revised version of the manuscript addresses all our previous concerns, and we find it is suitable for publication in its current form. We congratulate the authors for such a nice work.

Reviewer #3

(Remarks to the Author)

Point-by-point response letter

We would like to thank the Reviewers for their helpful comments and suggestions. We have studied these comments carefully and have made corrections accordingly in the revised manuscript and supplementary material (with changes marked in red font). Detailed point-by-point responses to the Reviewers' comments are given below.

We note that additional EDX, SEM, DSC/TGA, far-FIR, XRD and gas sorption measurements have been performed to address the Reviewers' comments. As such, we have included additional co-authors (Nina Lock, Ronghui Lu, and Rebekka Klemmt), who performed some of these experiments and assisted with the data analysis. All other co-authors have agreed with the inclusion of these additional co-authors.

Reviewer #1 (Remarks to the Author):

This is a nice paper of potentially broad interest across the field of glass science, materials engineering and beyond. The authors adopt empirical tools of classical glass chemistry in order to extend the range of MOF glass compositions – MOF glasses have emerged as a new class of glasses potentially combining the properties of metal-organic framework materials with the processing ability of glasses, but the range of MOF chemistries from which glasses can actually be produced has been limited to an extent which is currently preventing materials design through chemical formulation.

Response: We thank the Reviewer for their kind review of our manuscript and helpful comments below.

I have one sole concern with the paper, of which I am sure the authors can provide appropriate answers: in order to truly extend the range of available MOF glasses, the authors certainly agree that it is not sufficient to produce further glass-forming compounds which include a MOF as an ingredient, but which do not retain structural and physical properties of this MOF, in particular, its porosity. That said, a MOF glass is probably strictly only a MOF glass when it exhibits a metal-organic framework backbone.

According to the author's approach – with similarity to the classical network hypothesis developed for oxide glasses, in which the glass network can be "modified" using "network modifying species" – adding halide compounds to the MOF glass would reduce the network polymerization grade in similar ways as found in oxide glasses. In oxide glasses, the classification of what makes a network modifier is mostly empirical, being based on a lower threshold of oxide bond strength. This makes the current approach much more clear in that there is a clear-cut difference between the (MOF) network and the salt modifier.

That said, the 100% modified material in Fig. 1d should surely not be termed "ZIF", and it remains to be shown that the other two data points (50% and 25% modified) indeed retain a ZIF network topology.

Response: We appreciate the Reviewer's insightful comment regarding the similarity of our approach to the classical network hypothesis for oxide glasses as well as the importance of investigating the boundary between when the material constitutes a fully or partially polymerized metal-organic network vs. individual modified units. We also acknowledge the point that the material with 100% modifier content in Fig. 1d may no longer retain the typical characteristics of a ZIF network. As detailed in the manuscript (Results and Discussion, Structural Characterization section), the halide salts act as modifiers by replacing some of the organic linkers in the ZIF structure, leading

to depolymerization of the network. At high modifier concentrations, such as equal network to modifier ratio ($R=1$), the structure indeed transitions to a state where the connectivity between central zinc atoms is more or less equally shared by coordination to halide-anion as well as linkages through imidazolate. This has now been clarified in the revised manuscript (p. 16).

We note that in our structural analyses, particularly the X-ray pair distribution function (PDF) data (see Fig. 3c and Supplementary Fig. 32 in revised SI), the presence of Zn–N and Zn–Zn correlations even in the modified samples suggests that the primary network topology remains ZIF-like at lower modifier concentrations and that the Zn–N and Zn–Zn correlations persist even at a modifier content of $R=1$. Additionally, FT-IR and Raman spectroscopy results (Supplementary Figs. 15–16) confirm that the imidazolate linkers are retained in these compositions, albeit with partial replacement by halide anions. In addition, to address this and other comments, we have performed additional EDX measurements, which show no signs of inhomogeneity at the microscale (see Fig. 2d-h in the revised manuscript and **Response Fig. R1** below). This suggests that some ZIF-topology indeed remains in the obtained glasses. However, we do agree that it may not be meaningful to refer to especially the highly modified glasses as a ZIF. Therefore, to address this nomenclature issue in the revised manuscript, we have now referred to those materials with high modifier concentrations (e.g., $R=1$) as “halide-modified ZIF-derived glasses” rather than as “ZIF glasses” throughout the revised manuscript, emphasizing the significant structural transformation.

Response Figure R1. SEM images (leftmost column) and corresponding EDX data (remaining columns) of the prepared modified ZIF-derived glasses: (a) ZIF-4, (b) ZIF-62, and (c) ZIF-8 with bImCl, $R=1.0$.

Finally, we agree that our approach offers a more clear-cut differentiation between the network and the modifier species, as the halide salts are not integral to the original ZIF framework. This differs from the bridging and non-bridging terminology used for typical oxide glasses. We have included this point in the revised manuscript on p. 6 to highlight how our method builds upon the classical oxide glass modification framework while providing clarity in defining how network modification functions in the context of ZIF (and potentially other MOF) glasses.

I am not convinced that the NMR and PDF analyses presented in Fig. 3 provide such proof, in particular, as they are integral measurements from which it is not immediately clear whether the observed mid/longer-range features originate from a homogeneous material or from a (nano-)composite. But this puts into question the scheme proposed in Fig. 4. The same holds for the further short-range structural data provided in the SI.

Response: This is an important point about the interpretation of the NMR and PDF data and their relationship to the proposed mechanism in Fig. 4. We address these concerns as follows. Regarding the homogeneity of the modified ZIF-derived glasses, we agree that the nature of the NMR and PDF measurements makes it challenging to distinguish between whether our obtained materials are homogeneous and/or (nano-)composites. To better understand this, we have performed additional SEM and EDX analyses (**Response Figs. R1 and R2**). These show a uniform elemental distribution of Zn, Cl, N, and C in the modified materials, suggesting that no large-scale phase separation occurs. Our optical transparency tests of some of the modified ZIF-derived glasses (**Response Fig. R3** and in the revised Supplementary Fig. 10) also suggest high transparency (i.e., no scattering of particles with large size or different refractive index), indicating no significant variation in structural domains at the scale larger than a few hundred nanometers. We believe these observations support the nanoscale-homogeneity of the prepared samples. This data and analysis have been included on p. 8 of the revised manuscript and in Supplementary Figs. 9 and 10 in the revised SI.

Response Figure R2. EDX map sum spectra, showing the average elemental composition of the prepared modified ZIF-derived glasses: (a) ZIF-4, (b) ZIF-62, and (c) ZIF-8 with blmCl, $R=1.0$.

Response Figure. R3. Optical transmission data for ZIF-4-blmCl $R=1$ (as-cast and polished) and ZIF-62-blmCl $R=1$ (as-cast). Results are normalized to a sample thickness of 1.0 mm.

As a pragmatic solution, I would love to see physical property data, for example, glass adsorption, permeation and/or porosity analyses from which retained MOF porosity could be confirmed.

Response: We appreciate the Reviewer's suggestion. To investigate the porosity and surface area of the ZIF samples, we first conducted N_2 adsorption analysis at 77 K using the Autosorb iQ2 Automated Gas Sorption Analyzer (QuantaChrome). Upon initial testing, the ZIF-4 glass sample exhibited negative adsorption values in its isotherm (**Response Fig. R4a**). We verified the instrument's accuracy using Al_2O_3 (standard surface area: $5.29 \text{ m}^2 \text{ g}^{-1}$), which yielded a measured value of $5.34 \text{ m}^2 \text{ g}^{-1}$, confirming proper performance. We then analyzed another sample, ZIF-62-blmCl $R=1$ glass, which also displayed negative adsorption values (**Response Fig. R4b**). Based on the above SEM analysis, we conclude that these samples likely have low surface areas, and considerably lower than the Al_2O_3 standard.

Based on these findings, we shifted our focus to CO_2 adsorption experiments for a more accurate assessment of microporosity. In detail, we have conducted CO_2 adsorption-desorption measurements at $-42 \text{ }^\circ\text{C}$, $0 \text{ }^\circ\text{C}$, and $22 \text{ }^\circ\text{C}$ on selected modified ZIF-derived glasses. The results are presented in **Response Fig. R4c-i**. For all samples, CO_2 adsorption capacity increases as temperature decreases from room temperature to $-42 \text{ }^\circ\text{C}$. This is because a lower temperature reduces the molecular kinetic energy, enhancing adsorbate-surface interactions and favoring gas adsorption [see e.g. Raganati et al., Chem. Eng. J. 372, 526-535 (2019)]. Both ZIF-4 and ZIF-62 glasses (**Response Figs. R4c,d**) exhibit high CO_2 adsorption capacities with similar isotherm profiles, suggesting a dominant physisorption mechanism and the values of capacity are very close to those previously reported [Frentzel-Beyme et al., Nat. Commun. 13, 7750 (2022)]. However, the H_2 blmCl modification at $R=1$ significantly reduces the CO_2 adsorption capacity, likely due to pore collapse or altered surface chemistry, limiting adsorption sites.

Response Figure R4. (a) N_2 adsorption isotherm for ZIF-4 glass and (b) N_2 adsorption/desorption isotherm for ZIF-62 blmCI $R=1$ glass. (c-i) CO_2 adsorption isotherms for different samples under the temperature of $-42^\circ C$ (231 K), $0^\circ C$ (273 K), and $22^\circ C$, RT (295K): (c) ZIF-4 glass, (d) ZIF-62 glass, (e) ZIF-4-blmCI $R=1$ glass, (f) ZIF-62-blmCI $R=0.25$ glass, (g) ZIF-62-blmCI $R=1$ glass, (h) ZIF-8-blmCI $R=1$ glass, (i) ZIF-62-blmCI $R=1$ glass.

Interestingly, the ZIF-8-blmCI $R=1$ glass (**Response Fig. R4h**) exhibits superior CO_2 uptake compared to ZIF-4 and ZIF-62 glasses, indicating that framework-dependent effects influence adsorption behavior (e.g., ZIF-8 crystal is more porous than ZIF-4 and ZIF-62 crystals). A comparison between different functionalization levels reveals that ZIF-62-blmCI $R=0.25$ glass (**Response Fig. R4f**) exhibits lower adsorption than pristine ZIF-62, yet retain higher capacity than the fully modified $R=1$ counterpart (**Response Fig. R4g**). This suggests that partial incorporation ($R=0.25$) reduces pore disruption compared to full modification ($R=1$), leading to intermediate adsorption performance.

Furthermore, the CO_2 adsorption behavior of ZIF-62-blmCI $R=1$ glass (**Response Fig. R4i**) and ZIF-62 blmCI $R=1$ glass (**Response Fig. R4g**) exhibit a distinct temperature-dependent trend. At 295 K ($22^\circ C$), both materials show relatively low CO_2 uptake, with ZIF-62-blmCI $R=1$ displaying slightly lower adsorption than ZIF-62-blmCI $R=1$. However, as the temperature decreases to 273 K ($0^\circ C$) and further to 231 K ($-42^\circ C$), CO_2 uptake increases in both cases, with a more pronounced enhancement for ZIF-62-blmCI $R=1$. This suggests that the iodide-modified framework retains CO_2 more effectively at lower temperatures, potentially due to differences in surface affinity, structural flexibility, or kinetic

effects. Additionally, the larger size of the iodide ion may influence the free volume or pore accessibility of the glass, although further evidence is needed to confirm this effect. Interestingly, the increase in CO₂ adsorption capacity is more significant from 22 °C to 0 °C than from 0 °C to -42 °C for ZIF-62 blmI $R=1$. The findings highlight that while cooling enhances uptake by increasing adsorption strength, kinetic limitations (such as slower CO₂ diffusion or partial pore saturation) may begin to dominate at lower temperatures. These effects point to a combined influence of halide identity, thermodynamics, and mass transport in governing gas uptake performance. These results and analyses have been discussed on pp. 7-8 of the revised manuscript and the CO₂ adsorption measurements have been included as Supplementary Fig. 14 in the revised SI.

The sample photographs indicate relatively good material quality; I suggest that optical scattering spectra are provided on these samples in order to evaluate material homogeneity (vs. a potential composite nature) – Fig. S7 is perhaps a good start.

Response: We agree with the Reviewer that these additional data are helpful. In addition to the elemental composition analyses presented in **Response Figs. 1-2**, the optical transparency data are shown in **Response Fig. R3**. After polishing, the modified ZIF-derived glass surface exhibits numerous scratches, which adversely affect the light transmittance. In contrast, without polishing, ZIF-62-blmCl with $R=1$ achieves a maximum transmittance of 92.4%. Similarly, the transmittance of unpolished ZIF-4-blmCl reaches 85.5%, representing a significant improvement compared to the polished modified ZIF-derived glass. For a thickness of 1 mm, this is close to the transmittance of optical grade oxide glasses (albeit the wavelength window is somewhat smaller for the glasses studied herein). These results have been included as Supplementary Fig. 10 in the revised SI.

The R -parameter is not well defined. Perhaps it can be related to charge balance / charge compensation, or the Q-group speciation similar to the way this is done in oxide glass chemistry?

Response: We appreciate the Reviewer's suggestion regarding the definition of the R -parameter. We define the R -parameter as the molar ratio between the content of added modifier salt and the ZIF crystal, i.e., $R=n_{\text{modifier}}/n_{\text{ZIF}}$, where n_{modifier} and n_{ZIF} represent the amount of substance of the modifier and ZIF, respectively. While we acknowledge that other definitions could be used, it would then strictly relate to either a structural or chemical property. Conversely, we believe our definition based on the ratio of the modifier to the network former offers the simplest metric available. Such ratio of the modifier to the network former is also typically used in oxide glass chemistry [see, e.g., Jiusti et al., J. Non-Cryst. Solids 550, 120359 (2020)] to relate composition to various structural and physicochemical properties. Therefore, we prefer to maintain the R -parameter in this work.

I applaud the authors for the quality of their flash-DSC data! However, beyond Fig. S39 there is only little information on low-rate DSC, and there is no heat capacity data. If I understood correctly, all data provided are normalized; would it be possible to provide the non-normalized data, too?

Response: We address the specific points raised regarding low-rate DSC, heat capacity data, and non-normalized data in the following.

Low-rate DSC data. Our original manuscript relied on flash-DSC data due to its ability to resolve weak glass transitions and minimize structural relaxation effects during measurements. However, we acknowledge the value of including low-rate DSC data for comparison and broader applicability. Following the Reviewer's comment (and those of the other Reviewers below), we have conducted various low-rate DSC experiments to supplement the previously presented DSC data.

The new low-rate DSC data (measured at 10 K min⁻¹ in Al crucibles) are shown as **Response Fig. R5**. Specifically, in **Response Figs. R5a,b**, the first and second upscans of ZIF-62 and blmCl mixture with different molar ratio are shown, respectively. The strong endothermic peaks in panel (a) are not

associated with melting but with bursting of the crucible due to increased internal pressure and following release of mass from the sample (evaporation of trace solvents and free imidazole/benzimidazole species). Following the first upscan, the second scan shown in panel (b) reveals a clear glass transition peak. As the R -value increases (i.e., modifier content increases), T_g monotonically decreases from 60.8 °C to 31.8 °C. As shown in **Response Fig. R5c**, we have also performed measurements using different salts (benzimidazolium salts with Cl^- , Br^- , and I^- as counter anions) as modifiers in ZIF-62 crystals. The right-hand side of Response Fig. 5c presents the first heating scan, where the observed endothermic peaks again correspond to mass loss rather than melting, as indicated by the mass curve (dashed line). The left-hand side displays the results of the second heating scan, in which the mass remains constant. In this case, the endothermic peaks in the DSC curves correspond to the glass transition. T_g of ZIF-62-blmCl, ZIF-62-blmBr, and ZIF-62-blmI ($R = 1.0$) are found to be 29 °C, 27 °C, and 41 °C, respectively. Additionally, **Response Fig. R5d** illustrates the heating scan results for different crystals (ZIF-62, ZIF-4, and ZIF-8) modified with blmCl (at a ratio of $R=1$). The corresponding T_g values of the modified ZIF-derived glasses with different halide ions are 29 °C, 33 °C, and 30 °C, respectively.

Response Figure R5. DSC heating upscans at 10 K min⁻¹ in Al crucible. (a-b): First (a) and second (b) upscan for ZIF-62 and H₂blmCl with different R values (from 0.25 to 1.25). (c) First and second upscan for ZIF-62 with H₂blmCl, H₂blmBr, and H₂blmI for $R = 1.0$. (d) First and second upscan for ZIF-62, ZIF-4, and ZIF-8 with H₂blmCl for $R = 1.0$. The solid lines represent DSC (heat flow) data, and the dashed lines represent TGA data. Heat flow data have been shifted vertically for clarity.

Heat capacity data. We recognize the importance of isobaric heat capacity (C_p) measurements for a comprehensive understanding of the thermal properties of the modified ZIF-derived glasses. To complement our previous analyses, we have now performed standard DSC experiments in PtRh crucible at a heating and cooling rate of 10 K min^{-1} with a reference sapphire sample to determine the configurational heat capacity ΔC_p , i.e., the jump in isobaric heat capacity during the glass transition, for selected ZIF-blmCl samples. The results are shown in **Response Fig. R6**.

Response Figure R6. Isobaric heat capacity (C_p) measurements of (a) ZIF-8-blmCl for $R=1.0$, (b) ZIF-4-blmCl for $R=1.0$, and (c) ZIF-62-blmCl for $R=0.5$ and 1.0 . All data are collected at a heating and cooling rate of 10 K min^{-1} in Pt crucible.

For ZIF-8-blmCl sample with $R=1.0$ (**Response Fig. R6a**), a pronounced glass transition is observed at $\sim 50.8^\circ\text{C}$, with a ΔC_p of $\sim 0.25 \text{ J}\cdot\text{g}^{-1}\cdot\text{K}^{-1}$. While this value is higher than that reported previously for $a_g(\text{IL@ZIF-8-HT})$ ($0.11 \text{ J}\cdot\text{g}^{-1}\cdot\text{K}^{-1}$) [Nozari et al., *Nat. Commun.* 12, 5703 (2021)], it remains within the expected range for ZIF-derived glasses and reflects the influence of modifier incorporation on configurational entropy and network flexibility. The ΔC_p values for ZIF-4-blmCl $R=1.0$ (**Response Fig. R6b**) and ZIF-62-blmCl $R=1.0$ (**Response Fig. R6c**) are 0.28 and $0.39 \text{ J}\cdot\text{g}^{-1}\cdot\text{K}^{-1}$, respectively, which are significantly higher than those of unmodified ZIF-4 (0.11 and $0.16 \text{ J}\cdot\text{g}^{-1}\cdot\text{K}^{-1}$ for the LDA and HDA phases, respectively) and ZIF-62 ($0.19 \text{ J}\cdot\text{g}^{-1}\cdot\text{K}^{-1}$) [Bennett et al., *Nat. Commun.* 6, 8079 (2015); Qiao et al., *Sci. Adv.* 4, eaao6827 (2018)]. Interestingly, ZIF-62-blmCl $R=0.5$ (**Response Fig. R6c**) exhibits a ΔC_p of $0.22 \text{ J}\cdot\text{g}^{-1}\cdot\text{K}^{-1}$, which is lower than that of ZIF-62-blmCl $R=1.0$ and close to that of unmodified ZIF-62. These results suggest that increasing the modifier content leads to a higher degree of network depolymerization and configurational entropy, resulting in a more flexible and disordered glassy structure. The relatively lower ΔC_p of ZIF-62-blmCl $R=0.5$ compared to that of $R=1.0$ implies that at lower salt modifier concentrations, the network retains a higher degree of connectivity. This further supports the role of benzimidazolium chloride in its ability to tune the structural and thermal properties of the ZIF-derived glasses.

Finally, we generally acknowledge that the use of low-rate DSC is more common and provides a means of showcasing the possibility of preparing the sample under less extreme conditions than those using FDSC. As such, the new data show that the present materials can be prepared and studied using a standard low-rate DSC instrument. We have added discussion on this in the revised manuscript (p. 6-7) and included **Response Figs. R5** and **R6** as Supplementary Figs. 1 and 2 in the revised SI.

Non-normalized data. We note that only some of the flash DSC data are normalized and only when necessary to provide qualitative comparisons in plots. Specifically, all plots showing comparisons between different samples (e.g., Figure. 1c in the original manuscript) have data normalized to the maximum recorded heat flow of the scan to be able to include all the data using the same y-axis. Ideally, it would be better to normalize all scans by sample mass, but this is practically impossible due to the extremely small sample masses used in FDSC. However, when comparing measurements

of the exact same sample, we do not normalize and simply present the recorded heat flow in mW (see, e.g., Supplementary Fig. 7 in the revised SI). Measured heat flows in the range of 0.1-2 mW are common but are naturally very dependent on sample mass and heating rate. For the data currently normalized, we have opted to maintain this way of presenting the data but added a brief note about the general tendency of the heat flow in the Methods section (p. 21) of the revised manuscript.

A minor point: I do not agree that the MD simulations should be called “ab initio”, even when this is often done elsewhere.

Response: We have revised the manuscript and SI to refer to the simulations as “DFT-based MD” rather than “ab initio MD”. For example, see p. 15 of the revised manuscript.

Reviewer #2 (Remarks to the Author):

The manuscript by Sørensen, Smedskjaer and co-workers reports a new protocol to modify the melting of crystalline ZIFs by incorporation of additives, in a similar way to what is currently performed in oxide-based glasses. In order to prove this approach in ZIFs, the methodology has been generalized by using three different ZIFs (namely ZIF-4, ZIF-8 and ZIF-62), three different additives (benzimidazolium chloride, benzimidazolium bromide and benzimidazolium iodide) and different compositions ZIF:additive. This study is performed using Fast Differential Scanning Calorimetry (FDSC) in order to achieve the melting of the mixtures in a very short time, instead of using the common Differential Scanning Calorimetry (DSC) that has been typically used to report the melting of the ZIFs.

The idea of the work is interesting and could provide a way to modify the physical properties of this attractive type of materials. However, the use of the non-standard FDSC to produce the glassy form of the materials is not very convenient, as the results shown by the authors could be biased by this. Furthermore, as will be detailed below, the characterization of the materials does not show unequivocally the formation of the glass that the authors are proposing.

Response: We thank the Reviewer for their kind review of our work and helpful comments. Following the comments from all Reviewers, we have performed a variety of standard-rate DSC data, which clearly show evidence of glass formation. Please find our detailed response below as well as our response to the second-last comment of Reviewer #1 above.

1. The authors justify the choice of FDSC with two reasons (described in the Supplementary Text): the presence of weak glass transitions and the limitation of permanent structural changes. If a new methodology, as the proposed in this manuscript, is presented, standard systems should be used, not those with weak glass transitions. If the glass transitions change from clearly observable to weak, that could mean that the methodology is not suitable for this type of materials. In addition, the defense that FDSC is useful to limit permanent structural changes is again non-justifiable, as the new methodology should be valid in standard conditions (this should not be restricted to a heat speed of 500 K/s; if that is the case, then this work would not be of general interest).

Response: We choose to use FDSC due to the unique thermal behavior of the modified ZIF-derived glasses, which exhibit weak glass transitions due to their highly depolymerized nature and low liquid fragilities. We acknowledge that the preparation procedure using FDSC relies on extreme heating and cooling rates, and it is important, as pointed out by the Reviewers, to also showcase glass formation under standard conditions. However, we note that the investigated samples in the original manuscript were not directly prepared in the FDSC instrument, but rather only subsequently

analyzed using the FDSC method, i.e., the preparation procedure used a much lower cooling rate (see the “*Preparation of modified ZIF glasses*” section in the revised manuscript’s p. 20).

In any case, to address the Reviewer’s concern, we have supplemented the manuscript with additional measurements using a standard DSC for comparison (as shown in the **Response Fig. R5** above and related discussion). We both directly prepare the glass and followingly probe its glass transition using a standard rate of 10 K min⁻¹. We note that this was already briefly mentioned in the original manuscript (see Supplementary Fig. 39 in the original SI), but based on the newly measured data, we have significantly enhanced the discussion of this in the main text (p. 6-7) and provided additional new Supplementary Figs. 1 and 2 in the revised SI.

2. The characterization of T_g of the different samples has been performed using only one FDSC cycle. The T_g of some of these materials is rather unusual (e.g. 60 °C) and thus should be confirmed with additional DSC cycles. Again, the use of standard DSC instead of FDSC could facilitate this. In fact, the authors state, “We note how this melting temperature is below that of most other known ZIF systems,” and the result is compared with traditional ZIFs. However, it is misleading to draw conclusions from such a comparison, as FDSC is not directly comparable with conventional DSC.

Response: Again, we thank the Reviewer for raising the important point regarding the glass transition temperature (T_g) characterization and the comparability of FDSC results with traditional low-rate DSC. Indeed, T_g varies when measured at different scanning rates. Therefore, we have supplemented our data with standard DSC measurements at a conventional heating rate of 10 K min⁻¹ for representative modified samples. This allows us to compare the samples characterized using both standard DSC and FDSC. As shown in **Response Fig. R5** above, the T_g values of ZIF-62-blmCl $R=0.25$ and $R=1$ are approximately 29 and 61 °C, respectively. This is significantly below the values measured using FDSC (141 and 261°C, respectively, as presented in Table S1 in the revised SI using scanning rate of 500 K s⁻¹). While these differences are indeed very large, both measurement types provide clear glass transitions and thus show evidence of high glass-forming ability, even at standard rates. Finally, the large shift in T_g with heating rate can be explained by the very low liquid fragility as this is known to strongly influence the observed transition temperature with heating rate [for example, see Zheng et al., Chem. Rev. 119, 7848-7939 (2019)].

3. A fundamental technique to characterize the stability of the materials is Thermogravimetric Analysis (TGA), which provides information on the decomposition temperature. Are the materials stable enough to be heated at 500 K/s without any decomposition? Could the signal observed at 60°C be related with a partial decomposition of the ligands? In this sense, the black colour of the material resulting from heating ZIF-8 and the additives (e.g., Figure 2a, Supp. Fig. 2, or Supp. Fig. 10) suggests carbonization of the organic ligands. Thus, the nature of this solid is unclear.

Response: We thank the Reviewer for raising these important points regarding the thermal stability of the modified ZIF materials and the potential decomposition or carbonization of organic ligands during heating. To this end, we agree that TGA is a suitable technique to assess the thermal stability and decomposition behavior of these materials. While the original manuscript did not include TGA results, we have performed TGA analyses (as part of the low-rate DSC measurements mentioned above) for selected modified ZIF samples to address the Reviewer’s comments. As shown in **Response Figs. R5a,c,d**, during the first heating scan up to 210 °C, the sample mass curve generally begins to decline. This may be attributed to the partial release of imidazole/benzimidazole from the system after the re-polymerization of the salt and ZIF crystals (as noted in the modification mechanism description of Figure 4 in the main text). However, during the second upscan (**Response Figs. R5b,c,d**), the sample mass curve remains stable, indicating that no further mass loss occurs in this process.

Regarding the extreme heating/cooling rates experienced in FDSC (e.g., 500 K s⁻¹), this should be an advantage rather than a limitation in terms of thermal stability given that stability-limiting reactions will be kept to a minimum given the extremely short time the samples experience at high temperature. FDSC has, for example, previously been used to study otherwise unstable phases, which can only be stabilized using extreme cooling rates [see e.g. Kurtuldu & Loffler, Adv. Sci. 7, 1903544 (2020)].

Regarding the black colour of ZIF-8-blmCl/Br/I, we have found it to be unavoidable. In a previous study [Nozari et al., Nat. Commun. 12, 5703 (2021)], an ionic liquid (1-ethyl-3-methylimidazolium bis(trifluoromethanesulfonyl)imide) was incorporated into ZIF-8 to form a ZIF-8 based glass phase (IL@ZIF-8), which also exhibited a black coloration despite having a uniform structure at the microscale (see response to Reviewer #1 above and **Response Fig. R1** with new EDX measurements). We have not been able to unambiguously ascribe this coloration to a specific reaction but note that we have now confirmed the integrity of the organic linkers from the ZIF and salt modifier. Namely, as shown in **Response Fig. R7**, solution ¹H NMR spectra of a digested ZIF-8-blmCl R=1 glass sample reveal that the hydrogen signals on the methyl group (from methylimidazolate) remain unchanged. This suggests that the ligand structure remains intact without significant decomposition. Following the previous work [Nozari et al., Nat. Commun. 12, 5703 (2021)], we argue that during heating, salts may interact with the ligands of ZIF-8. In particular, at high temperatures, decomposition products of the modifier might react with ZIF-8 ligands, potentially leading to the dark coloration of the material. We have added a discussion on the general stability of the organic ligands in all the studied systems in the revised manuscript (p. 10-11) as well as included a new Supplementary Fig. 20 in the revised SI.

Response Figure R7. Solution ¹H NMR spectra of (a) ZIF-62 crystal, ZIF-62 glass, and ZIF-62-blmCl R=1, (b) ZIF-4 crystal and ZIF-4-blmCl R=1, and (c) ZIF-8 crystal and ZIF-8-blmCl R=1.

4. Solid state NMR has been used for the characterization of the solids, but the information that this technique provides is quite limited. In this case, the authors should provide liquid-state NMR of the

digested samples to clarify the ratio of imidazole (or methylimidazole) and benzimidazole incorporated in the final material for the different compositions.

Response: We agree that the interpretation of solid-state NMR is complex and needs supplement from, e.g., liquid state NMR of digested samples. As also mentioned above, we have therefore performed such measurements for ZIF-4, ZIF-8, and ZIF-62 derived glasses modified using H₂blmCl at $R=1$. As shown in **Response Fig. R7a**, the integration ratio of the hydrogen signals from benzimidazole (H 1#) and imidazole (H 2#) in the modified glass of ZIF-62-blmCl $R=1$ is 0.46:0.54, which is significantly higher than that observed in ZIF-62 crystal or glass (0.14:0.86) [Sørensen et al., *Chem. Mater.* 36, 2756-2766 (2024)] but close to the expected composition of ZIF-62-blmCl $R=1$ (0.42:0.58 from the expected composition of Zn(lm)_{1.75}(blm)_{1.25}Cl).

Additionally, characteristic benzimidazole peaks are present in the modified ZIF-derived glasses ZIF-4-blmCl $R=1$ (**Response Fig. R7b**) and ZIF-8-blmCl $R=1$ (**Response Fig. R7c**). The integration ratio of the hydrogen signals from benzimidazole (H 1#) and imidazole (H 2#) in ZIF-4-blmCl $R=1$ hybrid glasses is 0.38:0.62, which is relatively close to the expected composition ratio of 0.33:0.67 (expected composition is ZnIm_{2.0}blm_{1.0}Cl). The integration ratio of benzimidazole (H 1#) to methyl protons (H 6#) in ZIF-8-blmCl $R=1$ glass was determined as 0.34:0.66. After accounting for the three equivalent methyl groups in methylimidazole, this corresponds to an experimental benzimidazole-to-methylimidazole ligand ratio of 0.60:0.40. This result shows a deviation from the theoretically expected composition of Zn(mlm)_{2.0}(blm)_{1.0}Cl, which would expect a ligand ratio of 0.33:0.67. To address the Reviewer's comment in the revised manuscript, we have expanded the discussion of this in the main text (p. 10-11) and included solution-phase ¹H NMR spectra in Supplementary Fig. 20 in the revised SI.

5. The presence of Zn-Cl, Zn-Br, and Zn-I peaks in the PDF is quite revealing (Figure 3d) and provides strong evidence for the incorporation of halogens into the framework. However, EDX analysis should also be performed, in order to quantify the Zn:halide ratio of the final material. Is there a "ZnX₂N₂" unit, or is it a "ZnXN₃" unit? Or are there mixtures of all possible compositions? In addition, the electronic images are somewhat unclear, and it is difficult to visualize the distribution of atoms (Figure 2).

Response: We thank the Reviewer for recognizing the significance of our X-ray PDF measurements and we do agree that the EDX data could provide further knowledge on the sample compositions. As such, we have re-examined selected samples using SEM and EDX, as shown in **Response Figs. R1** and **R2** above, which now provide a clearer and more detailed visualization of the elemental distributions. Namely, the EDX results confirm the highly homogeneous distribution of key elements, including zinc (Zn), chlorine (Cl), carbon (C), and nitrogen (N), throughout the modified ZIF-derived glass matrices. The presence of chlorine (Cl) further supports the incorporation of blmCl into the modified ZIF-derived glass, while the Zn signal remains consistent with the expected framework composition. Generally, we find no evidence of inhomogeneities (e.g., phase separation) in the prepared glasses.

We also appreciate the Reviewer's suggestion to quantify the Zn:halide ratio to better understand the Zn coordination environment in the final material. To address this, we supplement the previous liquid ¹H NMR data with new EDX measurements to analyze the relative elemental compositions of C, N, Zn, and Cl in the modified ZIF-derived glasses. Based on these EDX results, the elemental weight fractions in the final material (taking ZIF-62-blmCl $R=1$ as an example) are summarized in **Response Table R1**. The calculated Cl/Zn ratio is 0.823:1, and the N/Zn ratio is 11.83:1, respectively (expected composition of ZIF-62-blmCl $R=1$ is ZnIm_{1.75}blm_{1.25}Cl: ZnC₁₄N₆H_{11.5}Cl). The Cl/Zn ratio is around 0.82, which suggests that each Zn is likely coordinated to approximately one halide ion on average, favoring a ZnX₁N₃ configuration. The same results for ZIF-4-blmCl $R=1$ and ZIF-8-blmCl $R=1$ are shown in **Response Tables R2** and **R3**, respectively.

We note that these calculations should only be considered as an approximation given the lack of suitable reference materials for the EDX calibration. The final composition of the modified ZIF-derived glass could be more complex than estimated here, and EDX alone is insufficient to provide precise compositional and structural data. As a comparison, we do for example note that traditional network glasses tend to feature distributions of structural units rather than one unique nanostructure entity. We have updated the SEM and EDX results in Fig. 2d-e in the revised manuscript and Supplementary Fig. 9 in the revised SI.

Response Table R1. Elemental distribution analysis using EDX of the obtained modified ZIF-derived glasses, synthesized on a hot plate for ZIF-62-blmCl $R=1$.

Element	Measured wt%	wt%*	Corresponding moles*	Expected wt%	Expected moles
C	50.2	52.9	24.40	47.6	14
N	28.4	30.0	11.83	23.8	6
Zn	11.2	11.8	1.0	18.5	1
Cl	5.0	5.3	0.823	10.0	1

* Proportion of each element among these four elements.; * Calculation is based on 1.0 mol of Zn.

Response Table R2. Elemental distribution analysis using EDX of the obtained modified ZIF-derived glasses, synthesized on a hot plate for ZIF-4-blmCl $R=1$.

Element	Measured wt%	wt%	Corresponding moles*	Expected wt%	Expected moles
C	49.0	50.2	24.69	45.8	13
N	32.8	33.6	14.17	24.6	6
Zn	10.8	11.0	1.0	19.2	1
Cl	5.1	5.2	0.87	10.4	1

Response Table R3. Elemental distribution analysis using EDX of the obtained modified ZIF-derived glasses, synthesized on a hot plate for ZIF-8-blmCl $R=1$.

Element	Measured wt%	wt%	Corresponding moles*	Expected wt%	Expected moles
C	51.0	53.6	18.75	49.4	15
N	21.5	22.6	6.77	23.0	6
Zn	14.8	15.5	1.0	17.9	1
Cl	7.9	8.3	0.984	9.7	1

6. It is unclear the final type of material that is formed in the glass. The additive that is used is composed of benzimidazole and halogen units. The halogen units are incorporated to the material, partly bound to Zn (although as stated in point 5, it is not clear how many halogens are bonded to Zn), and maybe part is uncoordinated (not revealed). What about the benzimidazole part? They could be bonded to Zn, or not.

Response: We thank the Reviewer for raising an important point regarding the final nature of the material. First, as discussed in our response to your comment #5 above, the EDX results confirm the incorporation of halogens, with a Cl/Zn ratio close to 1, suggesting that Zn is likely predominantly coordinated in a $ZnXN_3$ fashion (one halide per Zn, along with three N donors from imidazolate-type linkers). Pair distribution function (PDF) analysis (Fig. 3d in the revised manuscript) further supports Zn-halide bonding, as we find clear evidence of Zn-Cl, Zn-Br, and Zn-I bond lengths. We cannot

rule out that some halide species may remain uncoordinated or weakly associated within the glass, but we find a systematic increase in the halide-Zn correlation intensity of the PDF upon increasing the halide content (see Fig. 3c in the revised manuscript). This suggests a high degree of halide incorporation into the homogenous material.

The benzimidazole component (blm) in the benzimidazolium halide modifier can (i) remain as a free molecular species dispersed within the modified ZIF-derived glass, (ii) coordinate to Zn centers, replacing some of the original imidazolate linkers, and/or (iii) undergo chemical transformation due to thermal processing. Based on the existing solid-state ^{15}N and ^{13}C MAS NMR data (Supplementary Figs. 21 and 22 in the revised SI), we find chemical shifts consistent with free imidazole and benzimidazole species to be present in the spectra of the prepared modified ZIF-derived glasses. This suggests that the added benzimidazolium halide does indeed donate a proton to an imidazolate or benzimidazolate in the framework structure, producing one bonding and one free imidazole/benzimidazole. Furthermore, our new solution ^1H NMR spectra of the modified ZIF-derived glasses (see **Response Fig. R7**) indicate that benzimidazole retains its intact molecular structure, confirming that it does not undergo significant decomposition during the modification process. Moreover, we find a mass loss to take place from the sample at higher temperatures, which is caused by removal of both imidazole and benzimidazole (see Supplementary Fig. 41 in the revised SI). This suggests that (i) added benzimidazolium cations can replace existing organic linkers from the network, and (ii) imidazolate/benzimidazolate exists in a weakly bonded form (i.e., not directly bonding to Zn), which can easily escape from the homogenous structure.

Ultimately, the results confirm that halogens and benzimidazole are incorporated into the modified ZIF-derived glasses. EDX and PDF data suggest Zn coordination in a ZnXN_3 fashion (according to composition), and NMR results indicate proton transfer, forming both bonded and free benzimidazole species. Overall, these findings confirm a homogeneous, yet adaptable glass structure has been obtained based on our introduced approach, and we believe all of our results support the mechanism suggested in Figure 4 in the revised manuscript – although with the note that “A” in Figure 4 may in fact swap for the X-H structure, which coordinates to Zn. This is not shown directly in Figure 4, but to better reflect the opportunity for the ingoing imidazolium-species to replace the original linker when halide ions are added, we have added extra discussion about this in the revised manuscript (p. 16).

7. In Figure 1, the additives is described as “benzimidazolate halide” but this should be “benzimidazolium halide”. Also, the short version “benzimidazolium halide” could be clearer if H_2blmCl was used instead of blmCl .

Response: We thank the Reviewer for their careful reading. We acknowledge that the term “benzimidazolate halide” in Figure 1 is incorrect and should be revised to “benzimidazolium halide,” as the additive used is the protonated form of benzimidazole. In the revised manuscript, we have updated instances of “benzimidazolate halide” to “benzimidazolium halide,” including in Figure 1 and related text.

Furthermore, to enhance clarity and consistency, and to clearly distinguish between (benz)imidazolate, (benz)imidazole, and (benz)imidazolium, we agree with the suggestion to use H_2blmCl (benzimidazolium chloride) instead of the abbreviated form blmCl , where appropriate. However, in the final modified ZIF-derived glasses, the proton is not incorporated into the structure. Therefore, in naming the final glass compositions, we retain the blmCl notation (e.g., ZIF-62- blmCl $R=1.0$) to reflect the composition of the resulting material. Similarly, we have exchanged Im for imidazolium or imidazole where appropriate. We have applied these changes consistently throughout the revised manuscript, figures, and SI to ensure accurate chemical representation.

Reviewer #3 (Remarks to the Author):

Reviewer #4 (Remarks to the Author):

In this work, Cao et al. present a thorough and insightful study on the continuous structural modification of metal-organic framework (MOF) glasses via the introduction of halide salts as chemical modifiers. Through a combination of variable temperature X-ray total scattering and NMR data, the authors effectively demonstrate the coordination of the modifier with Zn^{2+} . This innovative method may broaden the landscape of MOF glass chemistry and may offer a platform for tuning the thermal and mechanical properties of these materials. However, some concerns remain regarding the thermal and structural analysis. In this regard, the manuscript may be considered for publication in Nature Communications after major revisions. The authors should consider addressing the following points to strengthen their work:

Response: We thank the Reviewer for their kind review of our work and helpful comments. We believe that we have addressed all the Reviewer's concerns point-by-point in the following.

1. In page 3, line 67, It would be beneficial to clarify that the pure phase of ZIF-76 cannot be melted (CrystEngComm, 2020, 22, 3627-3637; Angew. Chem. Int. Ed. 2024, 63, e202405307). In the original study, the melting observed was for a non-pure phase of ZIF-76 (Nat. Commun., 2018, 9, 5042). Additionally, the authors seem to ignore the fact that ZIF-8 is meltable (Nat. Commun., 2024, 15, 4420) after linkers exchange in their subsequent presentation of the effect of linker ratios on melting and the melting of ILs@ZIF-8.

Response: We appreciate the Reviewer's careful reading of our manuscript and the insightful suggestions for improvement. The work of Bumstead et al. [Bumstead et al., CrystEngComm 22, 3627-3637 (2020)] indeed states that melting and glass formation at ambient pressure do not occur in phase-pure ZIF-76 [Zn(Im)(ClIm)] or ZIF-76-mblm [Zn(Im)(mblm)]. This was our oversight, and we have revised the text accordingly in the revised manuscript.

The points we made regarding ZIF-8 not melting referred to the fact that under ambient pressure and direct heating, ZIF-8 undergoes decomposition before melting is observed [Gaillac et al., J. Phys. Chem. C 122, 6730-6736 (2018); Ma et al., Chem. Rev. 122, 4163-4203 (2022); Xue et al., Angew. Chem. Int. Ed. 63, e202405307 (2024)]. However, this does not contradict the fact that ZIF-8 can melt after linker exchange. To address these points, we have updated our Introduction section and added the relevant references in the revised manuscript.

2. It is noted that the glasses were prepared using a hotplate, which precluded the authors from capturing DSC signals during the first heating of the mixture. It would be good to measure the heat flow during the first heating using a closed aluminium crucible or high-pressure crucible to determine whether the modifiers melt individually or if there is a single co-melting endothermic signal. Additionally, could there be thermal behavior signal from the coordination of Zn^{2+} with halide ions? In other word, when would the reaction ZIF with halide salts occur, at the room-temperature ground preparation or the heating procedure of the mixture? The current understanding of the co-melting process, based only on variable temperature X-ray total scattering with full amorphisation around 300°C, is insufficient. Moreover, in Figure 1c, would it not be more appropriate to refer to a "supercooled liquid" rather than a "liquid"?

Response: We acknowledge the limitation of the methodology regarding the use of a hotplate, preventing us from recording the first heating DSC signals. This procedure was primarily employed to prepare larger samples than what is generally possible in a calorimeter (quantities in g vs. mg. respectively). However, to address the Reviewer's comment, we have performed additional low-rate DSC measurements to obtain heat flow curves at conventional heating rates (10 K min^{-1}) in closed Al crucibles. The results are shown in **Response Fig. R5** above. We note that a strong endotherm is observed during the first upscan of salt/ZIF-crystal mixture, which cannot be ascribed to melting but rather the bursting of the crucible as the internal pressure exceeds a few bars of pressure. This signal somehow obstructs weaker signals. Therefore, we have also conducted an additional experiment, where we used an Al crucible with a pinhole to allow continuous pressure release. This data is shown in **Response Fig. R8** below. From the first upscan of a mixture of ZIF-62-blmCl ($R=1$), we observe a small endotherm at around $100\text{ }^{\circ}\text{C}$, which we ascribe to the melting of pure benzimidazolium chloride. This is followed by a weak endotherm in the range of $200\text{-}250\text{ }^{\circ}\text{C}$, which we ascribe to the melting of the ZIF framework and homogenization of the sample. Next, the second upscan reveals a clear glass transition (red curve in **Response Fig. R8**). This appears consistent with the in-situ X-ray PDF data (Supplementary Fig. 32 in the revised SI), where amorphization is observed in the same temperature region. **Response Fig. R8** has been included as the new Supplementary Fig. 36 in the revised SI.

Response Figure R8. First and second DSC (solid lines) and TGA (dotted lines) heating upscans at 10 K min^{-1} in Al crucible (with a pinhole on the lid) of ZIF-62-blmCl $R=1.0$ sample.

The Reviewer also raises an excellent point regarding whether the reaction between the ZIF and halide salt occurs at room temperature during mixing or only upon heating. To study this, we have performed additional XRD analysis of the room-temperature mixture of salt and ZIF crystals. These data indicate that the crystal structure remains unchanged (see **Response Figs. R9a-c**, has been included Supplementary Fig.5 in the revised SI), suggesting that Zn-halide interactions primarily occur during elevated temperature and not at room temperature. We have also reevaluated the in-situ X-ray PDF data to further confirm this point. In detail, we here consider the heating data of ZIF-62-blmCl ($R=1$) sample, which is shown as **Response Fig. R9d** (and as Supplementary Fig. 26(b) in the revised SI). A clear appearance of the Zn-halide bond is first found to appear upon heating at around $100\text{ }^{\circ}\text{C}$. The intensity of this correlation increases with temperature up to around $200\text{ }^{\circ}\text{C}$ and

then stabilizes. We have added a brief discussion of the above points of reactivity in the revised manuscript (p. 7) and included the recorded XRD patterns as a new Supplementary Fig. 5 in the revised SI. We note that this data aligns very well with the observed calorimetry data in **Response Fig. 8**, which indicates the first endothermic reaction to occur at around 100 °C.

Response Figure 9. (a-c) XRD patterns of the mixture of salt (H_2bImCl) and crystal at room temperature for (a) ZIF-4- $bImCl$ $R=1.0$, (b) ZIF-8- $bImCl$ $R=1.0$, and (c) ZIF-62- $bImCl$ $R=1.0$. (d) In-situ PDF data of ZIF-62- $bImCl$ $R=1$.

Finally, the Reviewer suggests that in Figure 1c, the term "supercooled liquid" may be more appropriate than "liquid." We agree and have changed the terminology in the revised manuscript (p. 4) accordingly, as this phase exists below the equilibrium melting temperature and exhibits characteristics of an undercooled liquid rather than a fully equilibrated melt.

3. We also suggest measuring the change in heat capacity (ΔC_p), as this would provide insight into the degree of connectivity within the MOF glasses. How do the ΔC_p values of the modified ZIF glasses compare to those of pure ZIF-62 glass? It remains unclear whether this experiment can be conducted using FDSC, although the authors note that FDSC's fast heating rate allows the detection of weaker heat flow signals. Furthermore, how to define and/or differ the vitrification from melting for these ZIF- $bImCl$ $R=1$ cases in this study? Additional discussion of which may be beneficial and more clear for common readers.

Response: We thank the Reviewer for the good suggestion regarding the measurement of the jump in isobaric heat capacity during glass transition (ΔC_p) and its role in assessing the degree of connectivity within the modified ZIF-derived glasses. To this end, we have conducted standard DSC experiments at conventional heating and cooling rate (10 K min^{-1}) and successfully extracted ΔC_p values for the glass transition process. For the ZIF-8- $bImCl$ $R=1.0$ glass (see **Response Fig. R6a** above), a pronounced glass transition is observed at $\sim 50.8^\circ\text{C}$, with a ΔC_p of $\sim 0.25 \text{ J}\cdot\text{g}^{-1}\cdot\text{K}^{-1}$, a value comparable to that of other ZIF glasses, such as $a_9(IL@ZIF-8-HT)$ ($\Delta C_p=0.11 \text{ J}\cdot\text{g}^{-1}\cdot\text{K}^{-1}$) [(Nozari et al., Nat. Commun. 12, 5703 (2021))]. The ΔC_p values for ZIF-4- $bImCl$ $R=1.0$ (**Response Fig. R6b**) and ZIF-62- $bImCl$ $R=1.0$ (**Response Fig. R6c**) are 0.28 and $0.39 \text{ J}\cdot\text{g}^{-1}\cdot\text{K}^{-1}$, respectively, which are significantly higher than those of unmodified ZIF-4 (0.11 and $0.16 \text{ J}\cdot\text{g}^{-1}\cdot\text{K}^{-1}$ for the LDA and HDA

phases, respectively) and ZIF-62 ($0.19 \text{ J}\cdot\text{g}^{-1}\cdot\text{K}^{-1}$) [Bennett et al., *Nat. Commun.* 6, 8079 (2015); Qiao et al., *Sci. Adv.* 4, eaao6827 (2018)]. Interestingly, ZIF-62-blmCl $R=0.5$ (**Response Fig. R6c**) exhibits a ΔC_p of $0.22 \text{ J}\cdot\text{g}^{-1}\cdot\text{K}^{-1}$, which is lower than that of ZIF-62-blmCl $R=1.0$ and close to that of unmodified ZIF-62 ($0.19 \text{ J}\cdot\text{g}^{-1}\cdot\text{K}^{-1}$). These results suggest that increasing the modifier content leads to a higher degree of network depolymerization and configurational entropy, resulting in a more flexible and disordered glassy structure. The relatively lower ΔC_p of ZIF-62-blmCl $R=0.5$ compared to that of $R=1.0$ implies that at lower salt modifier concentrations, the network retains a higher degree of connectivity. This further supports the role of benzimidazolium chloride in tuning the structural and thermal properties of the ZIF-derived glasses.

Furthermore, as the Reviewer correctly points out, FDSC's fast heating rate enhances the detection of weak heat flow signals. However, it is generally not ideal for absolute C_p measurements due to the difficulties of accurately measuring the sample mass and thus establishing a reliable calibration of the heat flow. While our FDSC data provide insight into the relative changes in heat flow among the different samples, the standard low-rate DSC experiments obtained during the revision allow for a more quantitative evaluation of ΔC_p . We have clarified this in the revised manuscript, i.e., that the reported ΔC_p values are derived from standard DSC rather than FDSC (p. 7 in the revised manuscript).

Lastly, since the modified ZIF-derived glasses in our study were heated and formed on a hot plate, and their T_g values were measured using FDSC, this may have led to some confusion between the T_g and melting temperature (T_m). However, we believe this issue has now been clarified based on the performed standard DSC measurements conducted at a conventional heating rate (10 K min^{-1}), as shown in **Response Fig. R5d** above. In the first heating scan, an endothermic peak is observed, accompanied by mass loss in the TGA curve (**Response Fig. R8**). While such endotherm does not necessarily indicate a melting transition, it does suggest thermal events related to decomposition or some type of bond-breaking (intra- or interatomic). Based on the showcased stability of the organic linkers (see ^1H NMR data in **Response Fig. R7**), we argue that the endotherm is related to a mixture of evaporation of free imidazole/benzimidazole species and melting. However, in the second heating scan (**Response Fig. R8**), a clear endothermic peak is observed, while the TGA curve remains stable, indicating no further mass loss. Therefore, we infer that the transition around $60 \text{ }^\circ\text{C}$ corresponds to the glass transition.

4. In Fig. 3b, where VT-PDF data for ZIF-62-blmCl $R=0.5$ is presented, there are eight temperature points shown as S(Q) in Supplementary Fig. 18, yet only seven $G(r)$ data points are displayed in Fig. 3b. Additionally, the decision to switch from $R=1.0$ to $R=0.5$ in the main text appears to avoid the fact that $R=1$ is not fully amorphous (Supplementary Figure 24). This phenomenon is similarly observed in ZIF-62-blmBr $R=1$ (Supplementary Figure 28). In light of this, we recommend displaying $G(r)$ data up to 30 or 50 \AA to better assess the presence of long-range ordering. In the VT-PDFs, the Zn-Zn peaks at $\sim 6 \text{ \AA}$, often characteristic of ZIF glasses, are weak and asymmetric, raise doubts as to whether the final products are indeed ZIF glasses or coordination polymer (CP) glasses.

Response: We thank the Reviewer for these suggestions. In Figure 3b, there is indeed a missing temperature point in the image. This was a mistake, and we have corrected it in the revised manuscript.

The Reviewer suggests that the decision to present data for $R=0.5$ instead of $R=1.0$ in the main text may have been to avoid discussing the fact that $R=1.0$ is not fully amorphous. However, the primary reason for choosing $R=0.5$ was to facilitate comparison with the MD-simulated PDF results, which were difficult to conduct at higher concentrations due to space constraints in adding modifiers to the free space of the ZIF unit cell. As shown in Supplementary Fig. 3a in the revised SI, the XRD results confirm that at $300 \text{ }^\circ\text{C}$, blmCl-modified ZIF-62 undergoes complete amorphization when prepared

using our hot-plate method. However, in the in-situ PDF measurements at the synchrotron, the heating was conducted using a hot air flow, which differs from the hot-plate heating or DSC heating methods used in our laboratory. That is, the synchrotron heating setup featured significantly lower rates (and therefore much longer time at higher temperature). This difference in heating conditions have led to more time for evaporation of sample components, likely inducing some crystallization in selected samples. For instance, as shown in **Response Figs. 10a,b,c** below, at 300 °C, some crystalline peaks are still present. However, upon further heating to 330 °C, the samples become fully amorphous. We acknowledge that partial crystallization is present in the aforementioned samples and therefore we have added a brief discussion on this in the revised manuscript, including its likely cause of the low heating rate used at the synchrotron setup (pp. 22-23). In addition, we have added a plot of the PDF for the ZIF-62-blmCl ($R=1$) and ZIF-62-blmBr ($R=1$) samples, extending to a maximum radial distance of 30 Å. These have been included as a new Supplementary Fig. 47 in the revised SI. We note that the noisy spectra are due to the liquids shifting in and out of the beam during measurements, but we do not observe any clear structural order in the samples at length scales above 10 Å (in opposition to the crystals at lower temperatures).

Response Figure 10. Pair-distribution function $G(r)$ variable temperature data for (a) ZIF-62-blmCl $R=1$ (b) and ZIF-62-blmBr $R=1$ samples.

Regarding the Zn-Zn peaks observed at around ~ 6 Å in the PDF data, we agree that this peak is characteristic of ZIF crystals and glasses and their network structure. However, we note that the distinction between a ZIF glass and a coordination polymer glass is somewhat vague when going beyond the crystalline case and into the glassy state. As such, while the Zn-Zn correlation seems to weaken upon adding the modifying salt, it is retained in all the samples, indicating the continued connectivity of Zn-polyhedra within the structure. Given the retained distance, this link must be caused by the (benz/methyl)imidazolate linker, and from this point-of-view, the connectivity in the network mimics the connectivity in a crystalline ZIF. Reviewer #1 had a similar concern and to address this, we have in relevant places replaced the “ZIF-glass” terminology with “ZIF-derived glasses”. We have also added a brief discussion on the above points in the final discussion part of the revised manuscript (p. 14).

5. Page 12, line 283: The authors introduce 'imidazolium chloride' to examine the universality of their approach, yet the results appear divergent. In Supplementary Figure 31, PXRD data for both ZIF-8 and ZIF-4 reveal weak peaks around 15°, possibly attributable to the zni phase (ZIF-61 or ZIF-zni), for which no explanation is provided. Furthermore, no corresponding variable temperature X-ray total scattering data for ZIF-8-ImCl R=1 and ZIF-4-ImCl R=1 is offered. Also, the absent PXRD diffractions of ZIF-8-ImCl, ZIF-62-ImCl, ZIF-4-ImCl (R=1) before heating in Fig 1b, as well as supplementary Figure 1, make their unclear morphology or structure phase at room temperature. Are they glassy, gel or monolith?

Response: We appreciate the Reviewer's careful reading and observations regarding the divergence in results for imidazolium chloride (ImCl) modification and the lack of clear structural phase characterization before heating. First, the Reviewer notes the presence of weak peaks around 15° in the PXRD data for ZIF-8-ImCl and ZIF-4-ImCl, which could be attributed to the zni phase (ZIF-61 or ZIF-zni). Upon re-examination of the data, we recognize that these peaks likely correspond to residual crystallinity, rather than a fully glassy phase at intermediate temperatures. To address this, we have conducted additional higher-temperature synthesis experiments beyond 300 °C. The new XRD results on these samples confirm that when preparing them at higher temperatures (330 °C), as shown in the **Response Figs. R11a-c**, the peaks disappear, indicating full amorphization. These results have been included the new Supplementary Fig. 38 in the revised SI.

Response Figure R11. X-ray diffraction (XRD) patterns of ImCl modified (a) ZIF-62, (b) ZIF-4, and ZIF-8 derived glasses ($R=1.0$). The samples were prepared at either 300 or 330 °C). A comparison to pure imidazolium chloride (ImCl) is also included.

We also appreciate the Reviewer's suggestion to include variable temperature X-ray total scattering (VT-PDF) data for ZIF-8-ImCl ($R=1$) and ZIF-4-ImCl ($R=1$). While our primary focus was to demonstrate the melting point reduction and structural modification induced by benzimidazolium halide salts, we acknowledge that additional structural insights could be valuable. However, our main objective was to highlight the specific effects of H_2 ImCl on ZIF crystals. Although H_2 ImCl exhibits a similar effect in reducing the melting point, it is not the focus of our study. Moreover, the role of H_2 ImCl in lowering the melting point of ZIF crystals has already been clearly demonstrated through the PXRD data at different temperatures (**Response Fig. R11**), showing progressive structural changes with increasing temperature. The DSC measurements also confirm transitions induced by imidazolium chloride modification, which are similar to those of H_2 ImCl. Thus, while VT-PDF data for ZIF-8-ImCl and ZIF-4-ImCl could provide further details, we believe that the key impact of H_2 ImCl on melting behavior is already well-supported by our existing experimental data. Lastly, we note that additional VT-PDF would require a new synchrotron beamtime.

6. In Supplementary Figure 11, the peak at 421 cm^{-1} is unlikely to be due to Zn-N stretching. Zn-N stretching typically occurs in the far-infrared region around $\sim 300\text{ cm}^{-1}$ (Phys. Rev. Lett. 2014, 113, 215502). Far-infrared measurements are recommended to confirm the presence of Zn-N tetrahedra in the final glass. If absent, this would suggest the formation of a CP glass rather than a ZIF glass.

Response: We appreciate the Reviewer's insightful comment regarding the assignment of the 421 cm^{-1} peak in Supplementary Fig. 15 (in the revised SI) and the suggestion to conduct far-infrared (FIR) measurements to confirm the presence of Zn–N tetrahedra in the final glass. As suggested, we have thus performed far-infrared spectroscopy measurements (**Response Fig. R12**). Indeed, we can identify a relevant peak at about 310 cm^{-1} , which aligns well with the characteristic Zn–N stretching mode reported in the literature [Ryder et al., *Phys. Rev. Lett.* 113, 215502 (2014)]. This result supports the retention of Zn–N coordination in the glassy phase, indicating that the final material should be classified as a ZIF-derived glass rather than a coordination polymer (CP) glass. In addition, we identify a peak at around 290 cm^{-1} , which can likely be ascribed to features related to the incorporation of halide ions in the network as the band seems to be strongly influenced by the addition of the halide-containing crystal. We have discussed these new findings in the revised manuscript (p. 10), added new relevant references, and added the presented data as a new Supplementary Fig. 17 in the revised SI.

Response Figure R12. Far-infrared (FIR) measurements of (a) blmCl modified ZIF-62 derived glasses with different R values, (b) blmCl modified ZIF-62/4/8 derived glasses for $R=1.0$, and (c) ZIF-62 derived glasses modified by blmCl/Br/I ($R=1.0$).

7. Is the dark colour of the ZIF-8 glass due to the decomposition of the methyl group? Solution ^1H NMR of the digested samples is recommended for all samples, as this would provide valuable information on both the integrity of the linkers and the exact composition of the samples.

Response: We agree with this suggestion. As shown in **Response Fig. R7** above, the solution ^1H NMR spectra of ZIF-8-blmCl $R=1$ confirms that the hydrogen signals on the methyl group remain unchanged upon melting and quenching, suggesting that the vast majority of ligands remains intact during the glass formation. Based on the work of Nozari et al. [Nozari et al. *Nat. Commun.* 12, 5703 (2021)], we infer that during heating, salts may interact with the ligands of ZIF-8. In particular, at high temperatures, decomposition products of the modifier might react with ZIF-8 ligands, potentially leading to the dark coloration of the material. We refer the Reviewer to our response to Reviewer #2 above for a detailed description of the new liquid-state ^1H NMR findings. A discussion on the general stability of the organic ligands in all the studied systems has been included in the revised manuscript (p. 10).

8. FT-IR spectra (Supplementary Figure 11b) of ZIF-8-blmCl $R=1$ are markedly weakened and different as compared to others, which should be addressed and revised in corresponding main text, or be rerecorded. Additionally, we noted the binding energy for C 1s of XPS data (Supplementary Figure 11b) significantly shifted.

Response: We appreciate the Reviewer's careful reading and observations regarding the FT-IR spectra and shift in the C 1s binding energy in the XPS data. To further verify our findings, we have retested the FT-IR spectra for ZIF-8-blmCl $R=1$ and ZIF-8-blmCl $R=1.25$ samples, as shown in

Response Figure R13 below. These results confirm that both modified samples retain the same bonding characteristics as crystalline ZIF-8, indicating that no significant decomposition occurs at 300 °C. We have updated these findings in Supplementary Fig. 15 in the revised SI.

Response Figure R13. FT-IR transmission spectra of ZIF-8-blmCl ($R= 0.25\sim 1.25$) derived glasses.

Furthermore, as shown in **Response Figure R14** below, we have updated the XPS results for the modified ZIF-derived glasses. The C 1s XPS peaks in panels (a), (d), and (g) for blmCl-incorporated ZIF-derived glasses shift to a lower binding energy compared to crystalline ZIFs, indicating an increase in electron density around the carbon atoms. This shift is attributed to the trapping of π -electron-rich blm or lm species within the material, where their electronic interactions polarize the local environment, enhancing electron density within the carbon framework [Kümbetlioğlu et al., *ACS Omega* 8, 27650-27662 (2023)]. These data have been included in the new Supplementary Fig. 19 of the revised SI and discussed in the revised manuscript (p. 10).

Response Figure R14. (a-c) XPS data for (a) C 1s, (b) N 1s, and (c) Zn 2p of ZIF-4 and ZIF-4-bImCl R=1 glasses. (d-f) XPS data for (d) C 1s, (e) N 1s, and (f) Zn 2p of ZIF-8 and ZIF-8-bImCl R=1 glasses. (g-i) XPS data for (g) C 1s, (h) N 1s, and (i) Zn 2p of ZIF-62 and ZIF-62-bImCl R=1 glasses.

9. Confused label/names of glasses appeared in supplementary Figure 1, such as R=1 and R=1:1 shown, are they same as those in main text? Also, 'T_g of > 40K' in caption of Supplementary Figure 4 is not consistent with that shown in that plots.

Response: We thank the Reviewer for pointing out the inconsistencies in the labeling of glasses and the mismatch in data presentation in the supplementary figures. The label "R=1:1" in Supplementary Fig. 3 in the revised SI is indeed inconsistent with the labeling in the main text and other supplementary figures, where "R=1" is used. That is, we confirm that these labels referred to the same composition, and we have thus updated the labels in Supplementary Fig. 3 in the revised SI to ensure consistency with the main text and other figures. We also acknowledge that the "T_g of > 40K" phrasing was not suitable and well-explained, since the value corresponds to the full change of the T_g and that this was indeed not >40 K but rather ~30 K. We have corrected this in the caption of Supplementary Fig. 7 in the revised SI.

10. Through the author's experiments, it would be possible to remove additional modifiers (Nat. Commun.2024,15, 2040). Given the importance of porosity in ZIF glasses, it would be beneficial to provide CO₂ adsorption data at 195 K and the porosity information of the before/after modifiers removal.

Response: We appreciate the Reviewer's suggestion. To address this and a comment of Reviewer #1, we have now conducted CO₂ adsorption-desorption measurements on selected modified ZIF glasses. These results are presented in **Response Fig. R4** above. For all samples, CO₂ adsorption capacity increases as temperature decreases from room temperature to -42 °C. This is meaningful as lower temperature reduces molecular kinetic energy, enhancing adsorbate-surface interactions and favoring gas adsorption [Raganati et al., Chem. Eng. J. 372, 526-535 (2019)]. Both ZIF-4 and ZIF-62 derived glasses (**Response Figs. R4a,b**) exhibit high CO₂ adsorption capacities with similar isotherm profiles, suggesting a dominant physisorption mechanism. However, H₂blmCl modification at *R*=1 significantly reduces CO₂ adsorption, likely due to pore collapse or altered surface chemistry, limiting adsorption sites. Interestingly, the ZIF-8-blmCl *R*=1 derived glass (**Response Fig. R4f**) exhibits superior CO₂ uptake compared to ZIF-4- and ZIF-62-derived glasses, indicating that framework-dependent effects influence adsorption behavior. As such, we concluded that the modifier addition monotonously decreases the adsorption capacity of the systems, while the substitution of Cl⁻ for I⁻ induces no major change in capacity. We refer to our response to Reviewer #1 above for a more detailed description of the data. These results and analyses have been described on p. 8 of the revised manuscript and included as the new Supplementary Fig. 14 in the revised SI.

Finally, we acknowledge the Reviewer's point on the possibility to remove added modifiers, e.g., by heating or chemical leaching and that this might have a strong influence on the pore sizes and ultimately gas adsorption capacities of such materials. We agree that this could be of great interest but note that designing and understanding the chemistry behind this effect is outside the scope of the present study.

Reviewer #5 (Remarks to the Author):

Response Letter

Reviewer #1 (Remarks to the Author):

This has become a very nice manuscript. I greatly appreciate the way the authors handled my own and the other reviewer's comments and suggestions.

Response: We thank the Reviewer for their kind words and valuable suggestions on our work. We also thank the Reviewer for taking the time to go through our work again.

Reviewer #2 (Remarks to the Author):

The revised manuscript by Sørensen, Smedskjaer and co-workers has significantly improved with the new characterization techniques performed by the authors, which have served to clarify many of the points that were raised by the reviewers. Specially, one common point from all the reviewers was the lack of "standard DSC" measurements, which are now incorporated, and was one of the major criticisms.

However, the revised manuscript has still some issues that are not fully clear.

Response: We thank the Reviewer for acknowledging the improvements of the manuscript. We also thank the Reviewer for taking the time to go through our work again.

1. The composition of the glasses is not clear.

From the SEM-EDAX data shown in Supplementary Figure 9 (Response Figure R2), the Zn:Cl ratio is 2:1 for $R=1$, but this ratio should be 1, as $R = 1$ indicates that the number of moles of modifier and the number of moles of ZIF are the same (and there is 1 equivalent of Cl per modifier, and 1 equivalent of Zn per ZIF).

Response: This is an important observation. We would like to clarify that the data presented in Supplementary Figure 9 are based on weight fractions (wt%) as obtained from SEM-EDX analysis, and not directly on molar ratios. The wt% values of Zn and Cl reflect their relative mass contributions in the sample, which do not directly correspond to the stoichiometric molar ratio of Zn to Cl.

Given the significant difference in atomic masses between Zn (65.38 g/mol) and Cl (35.45 g/mol), a Zn:Cl molar ratio of approximately 1:1 would translate into a Zn:Cl wt% ratio close to 2:1, consistent with our obtained EDX results. Therefore, the observed wt% Zn:Cl ratio in the EDX spectra is fully consistent with the expected composition for $R=1$. In **Response Tables R1-3** (repeated below from the original Response Letter), we performed specific numerical calculations. It is important to note

that while EDX provides quantitative elemental data, its accuracy is limited for light elements such as carbon, nitrogen, and oxygen due to lower detection sensitivity, surface dependencies, and overlap with other elements in our EDX spectra. As such, the resulting weight fraction values should be regarded as approximate and as a guideline for assessing overall elemental composition rather than for precise stoichiometric quantification.

Response Table R1. Elemental distribution analysis using EDX of the obtained modified ZIF-derived glasses synthesized on a hot plate for ZIF-62-blmCl $R=1$.

Element	Measured wt%	wt%*	Corresponding moles*	Expected wt%	Expected moles
C	50.2	52.9	24.40	47.6	14
N	28.4	30.0	11.83	23.8	6
Zn	11.2	11.8	1.0	18.5	1
Cl	5.0	5.3	0.823	10.0	1

* Relative fraction of each element among these four elements.; * Calculation based on 1.0 mol of Zn.

Response Table R2. Elemental distribution analysis using EDX of the obtained modified ZIF-derived glasses synthesized on a hot plate for ZIF-4-blmCl $R=1$.

Element	Measured wt%	wt%*	Corresponding moles*	Expected wt%	Expected moles
C	49.0	50.2	24.69	45.8	13
N	32.8	33.6	14.17	24.6	6
Zn	10.8	11.0	1.0	19.2	1
Cl	5.1	5.2	0.87	10.4	1

* Relative fraction of each element among these four elements.; * Calculation based on 1.0 mol of Zn.

Response Table R3. Elemental distribution analysis using EDX of the obtained modified ZIF-derived glasses synthesized on a hot plate for ZIF-8-blmCl $R=1$.

Element	Measured wt%	wt%*	Corresponding moles*	Expected wt%	Expected moles
C	51.0	53.6	18.75	49.4	15
N	21.5	22.6	6.77	23.0	6
Zn	14.8	15.5	1.0	17.9	1
Cl	7.9	8.3	0.984	9.7	1

* Relative fraction of each element among these four elements.; * Calculation based on 1.0 mol of Zn.

The proposed formula is also wrong. In the response 5 to reviewer 2, where a detailed description of the composition of the different glasses is discussed, the formula used for this discussion is wrong: it is stated that the formula is $Zn(im)_{1.75}(blm)_{1.25}Cl$, but this formula indicates 4 negative charges and only 2 positive charges. This could be the result of part of the imidazoles/benzimidazoles being protonated and non-coordinated, e.g. something like $Zn(im)_{1.75}(bim)_{0.25}(H_2bimCl)$, which is charge-balanced.

Response: We thank the reviewer for their careful reading. We would like to clarify that the formula

provided in our earlier response (e.g., $\text{Zn}(\text{im})_{1.75}(\text{blm})_{1.25}\text{Cl}$) was indeed incorrectly quoted and should have read $\text{Zn}(\text{Im})_{1.75}(\text{blm})_{1.25}(\text{H}_2\text{blmCl})$. The latter corresponds to the previously quoted molar composition of $\text{ZnC}_{14}\text{N}_6\text{H}_{11.5}\text{Cl}$, which provides a charge-balanced state (assuming that nothing decomposes or evaporates during the melt-quenching process). We do, however, note that the two hydrogens attached in the H_2blmCl are altered during melt-quenching to provide charge-free blm and Im species in the final material. This was also noted in the previously submitted revised text in the discussion of the solid-state NMR measurements (and is now noted in the revised manuscript on p. 13).

The authors also mention that there is uncoordinated benzimidazole in the final material. However, it is not clear if this has been taken into account in the liquid NMR of digested samples.

Response: We confirm that the liquid ^1H NMR measurements were performed on fully digested samples, in which both coordinated and uncoordinated organic species are solubilized and analyzed together. As a result, the observed chemical shifts and integrations of the spectra reflect the total ligand population—including both Zn-bound and free (uncoordinated) benzimidazole and imidazole. While the liquid state NMR technique does not distinguish between coordinated and uncoordinated species directly, it remains highly valuable for assessing the overall chemical composition of the hybrid glasses. This highlights the importance of interpreting NMR results in conjunction with other techniques, such as TGA, PDF, and solid-state NMR, to collectively assess the coordination environment and structural disorder.

As shown in **Response Fig. R1a** and **Response Table R4** below (also shown as Supplementary Fig. 20 and Supplementary Table 4 in the revised SI), the integration ratio of the hydrogen signals from benzimidazole (H 1#) and imidazole (H 2#) in the modified glass of ZIF-62-blmCl $R=1$ is 0.46:0.54. This is significantly higher than that observed in ZIF-62 crystal or glass (0.14:0.86) [Sørensen et al., *Chem. Mater.* 36, 2756-2766 (2024)] but close to the expected composition of ZIF-62-blmCl $R=1$ (0.42:0.58 from the expected composition of $\text{Zn}(\text{Im})_{1.75}(\text{blm})_{1.25}\text{Cl}$).

Additionally, we note that characteristic benzimidazole peaks are present in the modified ZIF-derived glasses ZIF-4-blmCl $R=1$ (**Response Fig. R1b**) and ZIF-8-blmCl $R=1$ (**Response Fig. R1c**). The integration ratio of the hydrogen signals from benzimidazole (H 1#) and imidazole (H 2#) in ZIF-4-blmCl $R=1$ hybrid glasses is 0.38:0.62, which is relatively close to the expected composition ratio of 0.33:0.67 (expected composition is $\text{ZnIm}_{2.0}\text{blm}_{1.0}\text{Cl}$). The integration ratio of benzimidazole (H 1#) to methyl protons (H 6#) in ZIF-8-blmCl $R=1$ glass was determined as 0.19:0.81. After accounting for the three equivalent methyl groups in methylimidazole, this corresponds to an experimental benzimidazole-to-methylimidazole ligand ratio of 0.41:0.59. This result shows a slight deviation from the theoretically expected composition of $\text{Zn}(\text{mlm})_{2.0}(\text{blm})_{1.0}\text{Cl}$, which would expect a ligand ratio of 0.33:0.67. However, this discrepancy is not unexpected for ZIF-8-derived systems. In addition to

inherent uncertainties in EDX-based compositional analysis, methylimidazole (mIm) linkers are known to be thermally less stable and prone to partial decomposition—consistent with the fact that pure ZIF-8 does not form a glass on its own. In fact, the slight deviation of the benzimidazole : methylimidazole ratio may also be a sign of minor decomposition also related to the dark coloration observed in the ZIF-8-derived hybrid glasses.

Response Figure R1. Solution ¹H NMR spectra of (a) ZIF-62 crystal, ZIF-62 glass, and ZIF-62-blmCl R=1, (b) ZIF-4 crystal and ZIF-4-blmCl R=1, and (c) ZIF-8 crystal and ZIF-8-blmCl R=1.

Response Table R4. Peak areas and proton integration ratios from solution-state ^1H NMR spectra of digested samples. Proton peaks assigned to benzimidazole (blm) and imidazole (Im), or to methyl (CH_3) in the case of ZIF-8, were integrated to estimate the relative linker composition in each glass or crystalline sample. Expected ratios are given in brackets in the last column.

Sample	Peak of H 1# (blm)	Peak of H (H 2# Im or CH_3)	Integration ratio of H from blm: Im or CH_3
ZIF-62-blmCl R=1	1.0	1.16	0.46:0.64 (0.42:0.58)
ZIF-4-blmCl R=1	1.0	1.66	0.38:0.62 (0.33:0.67)
ZIF-62 Crystal	1.0	6.04	0.14:0.86 (0.13:0.87)
ZIF-8-blmCl R=1	1.0	4.37 (CH_3)	0.19:0.81 (0.33:0.67)

In addition, the new TGA data (Supplementary Figure 36, Response Figure R8) shows a mass loss of 20% in the first scan. Could this be related with the uncoordinated ligands? Could this be the cause of the black colour observed in some samples?

Response: As for the mass loss observed in the TGA data (Supplementary Figure 36), the initial ~20% mass loss below 300 °C is indeed attributable to the release of weakly bound or uncoordinated organic species, including free benzimidazole and imidazole. We believe this mass loss corresponds to the escape of uncoordinated ligands from the system, as discussed in the revised manuscript (pp. 15-16) and shown in Supplementary Figure 41 of the revised SI.

However, we do not attribute the black coloration observed in the ZIF-8-blmCl/Br/I samples to this mass loss or to the presence of uncoordinated ligands. As addressed in our previous response letter to Reviewer #2, comment 3, the black color appears to be an inherent outcome of the modification of ZIF-8 and is unavoidable under the current processing conditions. A similar observation was made in a previous study [Nozari et al., Nat. Commun. 12, 5703 (2021)], where an ionic liquid (1-ethyl-3-methylimidazolium bis(trifluoromethanesulfonyl)imide) was incorporated into ZIF-8 to form a glassy phase (IL@ZIF-8). This sample also exhibited a black appearance despite being microscopically uniform.

In contrast, the halide-modified ZIF-4-blmCl/Br/I and ZIF-62-blmCl/Br/I samples do not show black coloration under similar conditions. We speculate that this difference may arise from the 2-methylimidazolate linker in ZIF-8. The methyl group could potentially interact with Zn to form a dark-colored species, such as zinc complexes or carbon species from minor decomposition of the organic linker. Notably, even a small fraction of carbon species can give a strong dark color. While this is currently a hypothesis and beyond the scope of the current work, we plan to investigate this further in future studies.

2. New optical transmission data has been incorporated (Supplementary Figure 10, Response Figure R3), but the ZIF-8 derived glass has not been included here. This should be explained. Is it because

of the black colour?

Response: Indeed, the ZIF-8-derived glass was not included in the optical transmission data (Supplementary Figure 10) because it exhibits very low optical transparency, largely due to its black coloration. As a result, its transmission was below the detectable range of the instrument, and including the data would not provide meaningful insight. We acknowledge that this was not clear and have therefore clarified it in the newly revised manuscript (p. 8).

3. The new TGA data is not clearly presented: In Supplementary Figure 1 (Response Figure R5) this data is shown as dashed lines in a very small scale (right axis) which makes it extremely difficult to analyse. It seems there is some mass loss, but it's very difficult to tell. Also, in panels c and d of this figure, the data has a discontinuity at ca. 125 °C, which seems to be caused by the different scans in the DSC, but the values of the TGA do not seem to correspond to the end points of the first scans.

Response: Again, we thank the reviewer for their careful reading. In the original figure, we used dashed lines to present the TGA data to distinguish from the DSC signals. However, we acknowledge that this choice, along with the use of the secondary y-axis, made it difficult to interpret the TGA trends clearly. We have now revised the figure to improve clarity by adjusting the line style. The updated version has been included as Supplementary Figure 1 in the revised SI (as shown in the **Response Figure R2a,b** below).

Regarding panels c and d, we confirm that the same DSC heating/cooling program was used for all samples. That is, first a scan from -80 °C to 300 °C at 10 K min⁻¹, followed by cooling to -120 °C at 10 K min⁻¹, and then a second upscan to 120 °C at 10 K min⁻¹. To facilitate comparison, we plotted the two scans together in one panel and only included the most relevant portions of each. In the updated figure (**Response Figure R2c,d**), we have clearly distinguished the first and second scans using shaded backgrounds, with the first scan corresponding to the pink region on the right, and the second scan to the light-yellow region on the left. The associated TGA curves are shown in the upper portion of each panel.

Response Figure R2. DSC heating upscans at 10 K min^{-1} in Al crucibles. (a-b) First (a) and second (b) upscan for ZIF-62 and H_2blmCl with different R values (from 0.25 to 1.25). (c) First and second upscan for ZIF-62 with H_2blmCl , H_2blmBr , and H_2blmI for $R = 1.0$. (d) First and second upscan for ZIF-62, ZIF-4, and ZIF-8 with H_2blmCl for $R = 1.0$. The lower lines represent DSC (heat flow) data, while the upper lines represent TGA data. Heat flow data have been shifted vertically for clarity. Note that a small data jump is observed for the ZIF-62-blmCl $R = 0.25$ sample in panel (a), which is likely an experimental artifact and not a genuine thermal signal from the sample.

Regarding the discontinuity and difference in the mass signal from the end of the first upscan to the initiation of the second upscan, we acknowledge that these values are not similar. This is caused by a minor mass loss happening during the cooling process from $300 \text{ }^\circ\text{C}$ to $-120 \text{ }^\circ\text{C}$ at 10 K min^{-1} , which is not presented in the shown figures. We present an example of the full curve including up and downscans (as a function of time instead of temperature) in **Response Figure R3** below. We note that the heat flow signal during the cooling (downscan) is affected by fluctuations in the cooling rate, which leads to reduced resolution and less reliable thermal signal detection in this segment. As such, we have chosen to not include this data in the revised manuscript, but have added a comment in the revised Supplementary Information on the small difference in the mass from the end of the first upscan to the start of the second upscan (see caption of Supplementary Figure 1).

Response Figure R3. Full DSC curve including both heating (upscan) and cooling (downscan) segments plotted as a function of time, rather than temperature, for ZIF-62 modified with H₂blmCl at R = 1. The upscans were recorded at a heating rate of 10 K min⁻¹ in an Al crucible.

4. New adsorption isotherms (N₂ and CO₂) have been performed to study the porosity of the materials. The CO₂ sorption is presented for different materials in Supplementary Figure 14 (Response Figure R4), but the data is presented with different scales for the different solids. The unmodified glasses (panels c and d) show an uptake of 1.6 and 1.4 mmol CO₂/g, whereas the modified glasses novel in this study show an uptake of 0.03 mmol CO₂/g (panel e), 0.4 (panel f), 0.03 (panel g), 0.13 (panel h) and 0.04 (panel i), much lower than the original materials. In other words, the porosity is being lost. This figure should also be corrected, as Response Figure R4 has panels h and i which are missing in Supplementary Figure 14.

Response: Regarding the observed reduction in CO₂ uptake in the modified glasses, we agree with the Reviewer's conclusion: the porosity is significantly reduced upon incorporation of halide modifiers. This is consistent with our structural analyses, showing partial depolymerization of the ZIF framework and the presence of uncoordinated or loosely bound linkers, resulting in a decrease of accessible microporosity. We also acknowledge the Reviewer's point regarding the presentation of isotherms. While we attempted to unify the y-axes across all samples, this would cause the CO₂ uptake curves for the modified hybrid glasses to appear extremely compressed and nearly indistinguishable as shown in **Response Figure R4** below (the dataset corresponds to that in the revised Supplementary

Figure 14 with all panels plotted using a unified y-axis scale). For this reason, we maintained individual axis scaling for each panel to preserve data readability and clarity. However, we added a note to the caption of Supplementary Figure 14 to remind the reader that the y-axis range differs between the different samples.

Response Figure R4. CO₂ adsorption isotherms for selected samples at temperatures of 231 K (-42°C), 273 K (0°C), and 295 K (22°C, RT): (a) ZIF-4 glass, (b) ZIF-62 glass, (c) ZIF-4-blmCl R=1 glass, (d) ZIF-62-blmCl R=0.25 glass, (e) ZIF-62-blmCl R=1 glass, (f) ZIF-8-blmCl R=1 glass, and (g) ZIF-62-blmI R=1 glass. Note that the shown data are the same as in the revised Supplementary Figure 14, where all panels are plotted using a unified y-axis scale (maximum value: 2.0 mmol g⁻¹) to facilitate direct comparison of adsorption capacities across different samples. (h) N₂ adsorption isotherm for ZIF-4 glass and (i) N₂ adsorption/desorption isotherm for ZIF-62-blmCl R=1 glass.

Concerning the figure labeling, we note that the previous Response Figure R4 (i.e., in the initial resubmission) included additional panels that are not present in Supplementary Figure 14. Specifically, N₂ adsorption isotherm for ZIF-4 glass and N₂ adsorption/desorption isotherm for ZIF-62-blmCl R=1 glass. These additional datasets were obtained as part of our initial investigation using N₂ adsorption at 77 K, performed on an Autosorb iQ2 Automated Gas Sorption Analyzer (QuantaChrome). In these tests, the ZIF-4 glass sample showed negative adsorption values (shown

here above as **Response Figure R4h**). To verify the measurement accuracy, we analyzed a certified reference sample of Al₂O₃ (BET surface area: 5.29 ± 0.21 m²/g), which yielded a measured surface area of 5.34 m²/g, confirming proper instrument performance. We then tested the ZIF-62-blmCl R=1 sample, which also produced negative adsorption values (presented above as **Response Figure R4i**). Based on these results and our SEM observations, we conclude that the modified ZIF-derived glasses possess very low surface areas, significantly lower than the Al₂O₃ standard. This finding motivated our decision to focus on CO₂ adsorption measurements, which are more sensitive to small pore volumes and better suited for probing low-porosity materials. As such, since the obtained dataset using N₂ is not meaningful, we have omitted these adsorption isotherms from the manuscript. However, in the newly revised manuscript, we have added a small discussion about the performed N₂-adsorption measurements at 77 K, concluding that these did not yield meaningful data due to low surface areas (see p. 9 in the revised manuscript). We believe this text helps to clarify our reason for using CO₂ instead of N₂ as the adsorption gas.

5. The compositional mapping from the SEM images should present a clear view of the solid with its edges, so it can be clearly observed the presence of the elements in the material and the absence of the elements where no material is present. The current images shown in Figure 2 (and in Response Figure R1) make it impossible to unambiguously determine the presence or absence of the elements.

Response: We acknowledge that the elemental maps shown in Figure 2 in our initial revised manuscript (and Response Figure R1 in the first response letter) do not clearly delineate the edges of the solid, making it difficult to unambiguously determine the presence or absence of elements relative to the material boundaries. To address this concern, we have supplemented the elemental mapping data with the corresponding sum spectra of the EDX maps (see **Response Figure R5** below). These sum spectra clearly show the presence of the expected elements (C, N, Cl, Zn), as well as Pt from the conductive coating, and importantly, the absence of major unexpected elements, as no additional peaks are present. To improve readability, we have replotted these spectra and labeled the elemental peaks more clearly. In addition, we have added a new Supplementary Table 3 in the revised SI (shown as **Response Table R5** below) that lists the quantified elemental compositions (in wt%) derived from the EDX sum spectra using the AzTecLive software. This allows for a clearer interpretation of the compositional results beyond the maps alone. Furthermore, we have included the following clarification in the revised Supplementary Text (page 3):

“The sum spectra of the EDX maps (Supplementary Fig. 9) show distinct peaks for C, N, Zn, Cl, and Pt (from coating), confirming the presence of these elements and the absence of other elemental contributions. The quantified compositions obtained from the sum spectra are summarized in Supplementary Table S3.”

Finally, while we acknowledge that the original compositional maps alone may not fully resolve the element distribution near the sample edges, we believe the revised presentation - combining clear sum spectra and quantitative analysis –confirms the uniform presence of elements such as the halide modifiers in the glass matrix.

Response Figure R5. EDX sum spectra for the modified ZIF-derived glasses with H_2blmCl at $R = 1$ for (a) ZIF-4-blmCl, (b) ZIF-8-blmCl, and (c) ZIF-62-blmCl. Characteristic peaks corresponding to C, N, O, Cl, Zn, and Pt (from conductive coating) are indicated.

Response Table R5. Elemental composition of the modified ZIF-derived glass (in wt%) and standard deviation as determined from the fit of the sum spectra by the AztTeclive Standard software.

	C (wt%)	N (wt%)	Zn (wt%)	Cl (wt%)	O (wt%)	Na (wt%)
ZIF-4-bImCl R=1.0	49.0±0.2	32.8±0.2	10.8±0.0	5.1±0.0	2.3±0.1	0.0±0.0
ZIF-8-bImCl R=1.0	50.2±0.2	28.4±0.2	11.2±0.1	5.0±0.0	3.0±0.1	2.2±0.0
ZIF-62-bImCl R=1.0	51.0±0.2	21.5±0.3	14.8±0.1	7.9±0.0	2.7±0.1	2.2±0.0

6. The temperature dependent studies are shown in the different figures with the temperature indicated in the right hand side of the figure, in a graph. This is a nice way of presenting the data, but the authors might consider aligning the temperature points with the corresponding graphs of the left part, so it would be much easier the identification of the temperature (in addition to the different colour used)

Response: We thank the Reviewer for their constructive suggestion. We agree that aligning the temperature points more precisely with the corresponding graphs on the left would improve the readability and facilitate easier identification of the measurement temperatures, in addition to the use of different colors. As such, we have adjusted the layout of the temperature markers in the revised figures (Figure 3b in the revised manuscript and Supplementary Figures 24-35, 37, and 45 in the revised SI). We believe these modifications enhance the clarity of the presentation.

Reviewer #3 (Remarks to the Author):

Reviewer #4 (Remarks to the Author):

The authors have made a commendable effort in revising the manuscript. They have conducted and included a substantial amount of additional experimental work, which has addressed nearly all of the concerns. The new data significantly strengthen the conclusions of the study and enhance its overall scientific value.

However, Figure 3b seems to have forgotten to be updated to 8 Gr data. With this minor correction, we believe the manuscript meets the high standards of Nature Communications.

Response: We thank the Reviewer for acknowledging the improvements of the manuscript. We also thank the Reviewer for taking the time to go through our work again.

We acknowledge that the update to Figure 3b was inadvertently overlooked in the previous submission. This has now been corrected in the revised version, with eight $G(r)$ datasets now being properly included.

Reviewer #5 (Remarks to the Author):
